# Increases in reef size, habitat and meta-community complexity associated with Cambrian radiation oxygenation pulses

Andrey Yu. Zhuravlev [1] ✉, Emily G. Mitchell [2] ✉, Fred Bowyer [3], Rachel Wood [3] & Amelia Penny [4]

Oxygenation during the Cambrian Radiation progressed via a series of short-lived pulses. However, the metazoan biotic response to this episodic oxygenation has not been quantified, nor have the causal evolutionary processes been constrained. Here we present ecological analyses of Cambrian archaeocyath sponge reef communities on the Siberian Platform (525–514 Ma). During the oxic pulse at ~521–519 Ma, we quantify reef habitat expansion coupled to an increase in reef size and metacommunity complexity, from individual within-community reactions to their local environment, to ecologically complex synchronous community-wide response, accompanied by an increase in rates of origination. Subsequently, reef and archaeocyath body size are reduced in association with increased rates of extinction due to inferred expanded marine anoxia (~519–516.5 Ma). A later oxic pulse at ~515 Ma shows further reef habitat expansion, increased archaeocyath body size and diversity, but weaker community-wide environmental responses. These metrics confirm that oxygenation events created temporary pulses of evolutionary diversification and enhanced ecosystem complexity, potentially via the expansion of habitable space, and increased archaeocyath individual and reef longevity in turn leading to niche differentiation. Most notably, we show that progression towards increasing biodiversity and ecosystem complexity was episodic and discontinuous, rather than linear, during the Cambrian Radiation.

The rise of animals (metazoans) is a seminal event in the history of life. The Cambrian Radiation ~540 Ma marks the appearance of abundant and diverse metazoans and increasing ecosystem complexity in the fossil record[1]. A causal relationship between the redox and fossil records is proposed, where oxygen provision reached a threshold, or series of thresholds, which allowed for the diversification of metazoans with increasing metabolic demands[2]. Global geochemical data, however, suggest that oxygenation was not a simple, linear process, but rather occurred episodically via a series of short-lived pulses (1–3 Myr), or 'oceanic oxygenation events' (OOEs)[3,4]. Early and even later Cambrian seas likely had shallower, and more dynamic, oxygen minimum zones (OMZs) than modern oceans[5,6]. Such pulses of increased oxygenation (and related changes in productivity) are hypothesised to have increased the extent of shallow-ocean oxygenation and hence to have promoted diversification[7]. But what remains unquantified is the community-wide response of metazoans to such redox cycles, an insight into the evolutionary processes involved, and hence whether these pulses were indeed a driving force for the Cambrian Radiation.

[1]Department of Biological Evolution, Faculty of Biology, Lomonosov Moscow State University Leninskie Gory 1(12), Moscow 119234, Russia. [2]Department of Zoology, University of Cambridge, Museum of Zoology, Downing Street, Cambridge CB2 3EJ, UK. [3]School of GeoSciences, University of Edinburgh, James Hutton Road, Edinburgh EH9 3FE, UK. [4]Centre for Biological Diversity, School of Biology, University of St Andrews, Greenside Place, St Andrews KY16 9TH, UK. ✉e-mail: ayzhur@mail.ru; ek338@cam.ac.uk

In order to test the hypothesis that oxic pulses led to diversification and potentially ecological development, a correlation between increased oxygenation, rates of origination, and metrics of metazoan ecosystem complexity needs to be demonstrated. Early Cambrian marine environments were heterogeneous with respect to oxygen provision and nutrient load at a regional scale, so in order to investigate potential correlations, we require the integration of global and local redox proxies, and biotic records in the same stratigraphically well-constrained geological successions.

During the early Cambrian, the Siberian Platform was a vast isolated, tropical continent almost entirely covered by an epicontinental sea (Fig. 1a)[8,9]. The platform supported a single metacommunity, i.e. a species pool with many local, interacting communities e.g.[10], representing a third of total early Cambrian metazoan benthic diversity with widespread metazoan (archaeocyath sponge) reefs that formed bioherms (Fig. 1b)[7,11]. Dynamic and synchronous changes of body size in archaeocyath sponges, hyoliths, and helcionelloid molluscs through the early Cambrian on the Siberian Platform have been quantified, which coincide with elevated biodiversity and rates of origination: these have been proposed to follow OOEs[12]. Here we consider temporal changes in both the position of archaeocyath sponge reefs as a function of relative water depth, and in individual reef size (diameter), as well as the ecological complexity of the reef-building and dwelling communities by quantification of changing reef community membership of sessile archaeocyath sponge, coralomorph, and cribricyath species, on the Siberian Platform.

To quantify ecological complexity, we used metacommunity analyses, which compare the structure between communities in terms of taxa (generally species) compositions spatially and temporally[10] (see Methods). The 'Elements of Metacommunity Structure' framework used here is a hierarchical analysis that identifies properties in site-by-species presence/absence matrices that are related to the underlying processes, such as species interactions, dispersal, and environmental filtering that shape species distributions[10]. Application to various marine and terrestrial palaeocommunities has demonstrated the robustness of these methods to fossil data and sample size variations[13,14]. There are fourteen different types of metacommunity structure which are determined by the calculation of three metacommunity metrics: Coherence, Turnover, and Boundary Clumping, which reveal different controlling processes of underlying metacommunity structure[10,15–18].

The most ecologically complex metacommunities are classified as Clementsian, and have positive coherence, turnover and boundary clumping[16]. Clementsian metacommunities contain groups of taxa with similar range boundaries that respond to the environment synchronously as taxa have physiological or evolutionary trade-offs within the communities associated with environmental thresholds[19]. By contrast, when taxa respond individualistically to the underlying environment, without accounting for other taxa within the community, the structure is Gleasonian, and is defined by positive coherence and turnover but no significant boundary clumping[16]. When coherence is positive, but turnover is not significantly different from random, then the resultant metacommunity structures are known as quasi-structures (e.g. quasi-Clementsian), which reflect weaker underlying structuring processes.

We determined the metacommunity structure for archaeocyath sponge species on the Siberian Platform throughout their early Cambrian record using an entire previously published data set[11] then on a sub-set of metacommunities which had a sufficient number of reef sites to be suitable for analyses, i.e. with a sufficient number of sites to be statistically significant. Further, to investigate the effects of water depth on metacommunity structure, we used Spearman rank correlations to test whether the metacommunity ranking (as determined by reciprocal averaging, a type of correspondence analysis which ordinates the sites based on their species composition[17]), is significantly correlated to water depth. Finally, to quantify how pairwise associations between taxa change between the three temporally different metacommunities, we determined which pairwise taxa co-occurrences are significantly non-random using a combinatorics approach, and whether any non-random co-occurrences are positive or negative[20].

Species richness estimates are highly sensitive to differences in sampling. When comparing species richness of assemblages from several time intervals, it is advisable to standardise sampling across those assemblages to ensure that changes in species richness are not attributable to sampling differences. One approach is to subsample each time interval down to a standardised number of individuals (size-based rarefaction), but this approach can underestimate changes in richness because it tends to sample low-richness assemblages more completely than high-richness ones[21]. Coverage-based rarefaction, where each sample is down-sampled to a standardised level of taxonomic completeness, avoids this potential issue. The coverage of a sample is the proportion of species in the assemblage which are represented in that sample, and it can be estimated by subtracting the proportion of singletons in a sample from 1 (e.g.[22]; see also[21] for details). We used the estimateD function from R package iNEXT[23] to produce coverage-standardised species richness estimates with 95% confidence intervals, by repeatedly down-sampling the sampled assemblage from each time interval to match the coverage of the lowest-coverage interval. We did this by setting datatype = "abundance", base = "coverage" and leaving all other arguments as default.

In sum, we test the biotic response to OOEs by compiling metrics of archaeocyath reef size, location, and metacommunity complexity, integrated with existing data on archaeocyath individual size, species richness and origination and extinction rates[12] and high-resolution geochemistry[4,7] recalculated to the same stratigraphic scale, on the Siberian Platform over 11 Myr through Cambrian stages 2–3 (mid-Tommotian to early Botoman on the Siberian stratigraphic scale; 525–514 Ma). These results are used to quantify the community-wide response of metazoans to extrinsic redox cycles, and hence gain insight into the evolutionary processes involved.

## Geological setting and evolution of redox

During the early Cambrian shallow marine carbonates associated with evaporites and siliciclastics dominated the inner Siberian Platform, passing to shallow marginal carbonates of transitional facies known as the transitional zone (or the Anabar-Sinsk), thence to deep ramp and slope settings that accumulated organic-rich limestone and shale (Fig. 1a)[24–26]. Archaeocyathan reefs or bioherms were almost entirely restricted to the transitional facies. Such reefs appeared and proliferated during Cambrian stages 2 and 3 (Tommotian, Atdabanian and earliest Botoman), disappeared at the beginning of Stage 4 (middle Botoman) and re-appeared briefly at the end of this stage (Toyonian).

We integrate palaeontological (archaeocyath species number and individual size), palaeoecological (reef size and palaeodepth location) and chemostratigraphic information (carbon isotope cycles 5p, 6p, and II–VII) for sections of the Aldan, Selinde and Lena rivers with sub-metre-scale lithostratigraphic subdivisions[27–33] (Figs. 1c, 2a–c, 3a). This results in negligible uncertainty associated with sample heights, which are fixed relative to a consistent datum within each section.

Throughout Cambrian stages 2 and 3, high-amplitude positive $\delta^{13}C$ carbon isotope excursions show a strong positive covariation with the sulphur isotope composition of carbonate-associated sulphate ($\delta^{34}S_{CAS}$) in sections from the Lena River (Fig. 3b)[7]. The rising limbs of these excursions are interpreted as intervals of progressive burial of reductants under anoxic bottom water conditions, and a progressive increase in atmospheric oxygen[7]. Coincident $\delta^{13}C$ and $\delta^{34}S_{CAS}$ peaks (numbered II–VII) correspond with a pulse of atmospheric oxygen into the shallow marine environment (creating an OOE), followed by a corresponding decrease in reductant burial under more widespread

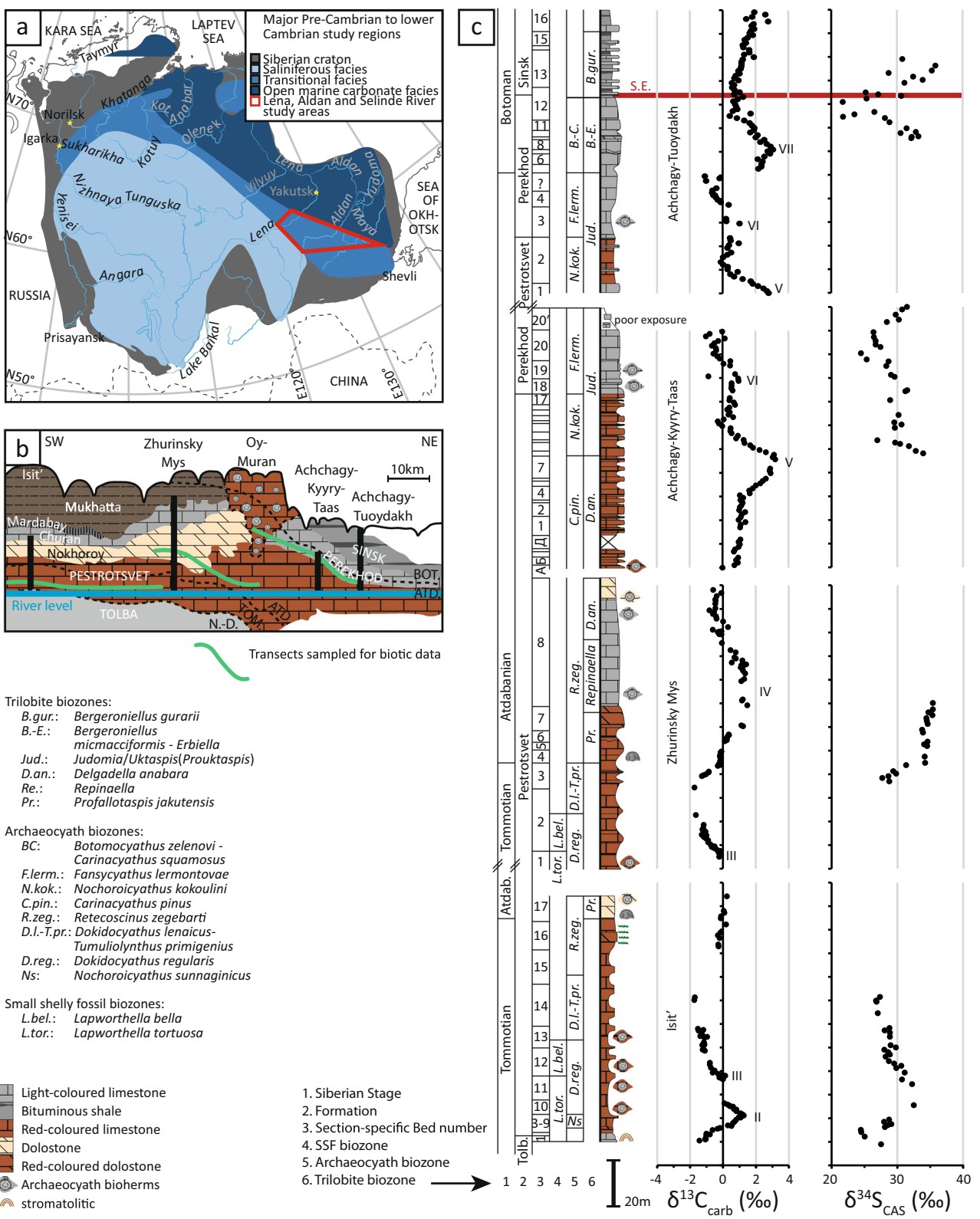

**Fig. 1 | Palaeogeographic and stratigraphic position of the early Cambrian archaeocyath reefs of the Lena-Aldan area on the Siberian Platform. a** Early Cambrian palaeofacies zonation map of the Siberian Platform. **b** Cross section to show relative positions of sampled transects along the Lena River[11,40,66–68]. **c** Lithostratigraphy, biostratigraphy, carbon isotope (δ13C)[29,31,32] and carbonate-associated sulfate sulfur isotope (δ34SCAS)[7] data for sections from the middle Lena River (Isit', Zhurinsky Mys, Achchagy-Kyyry-Taas, and Achchagy-Tuoydakh). S.E.−Sinsk Event; Tolb.−Tolba Formation; ATD., BOT., N.-D., TOM.−Atdabanian, Botoman, Nemakit-Daldynian, and Tommotian local stages, respectively.

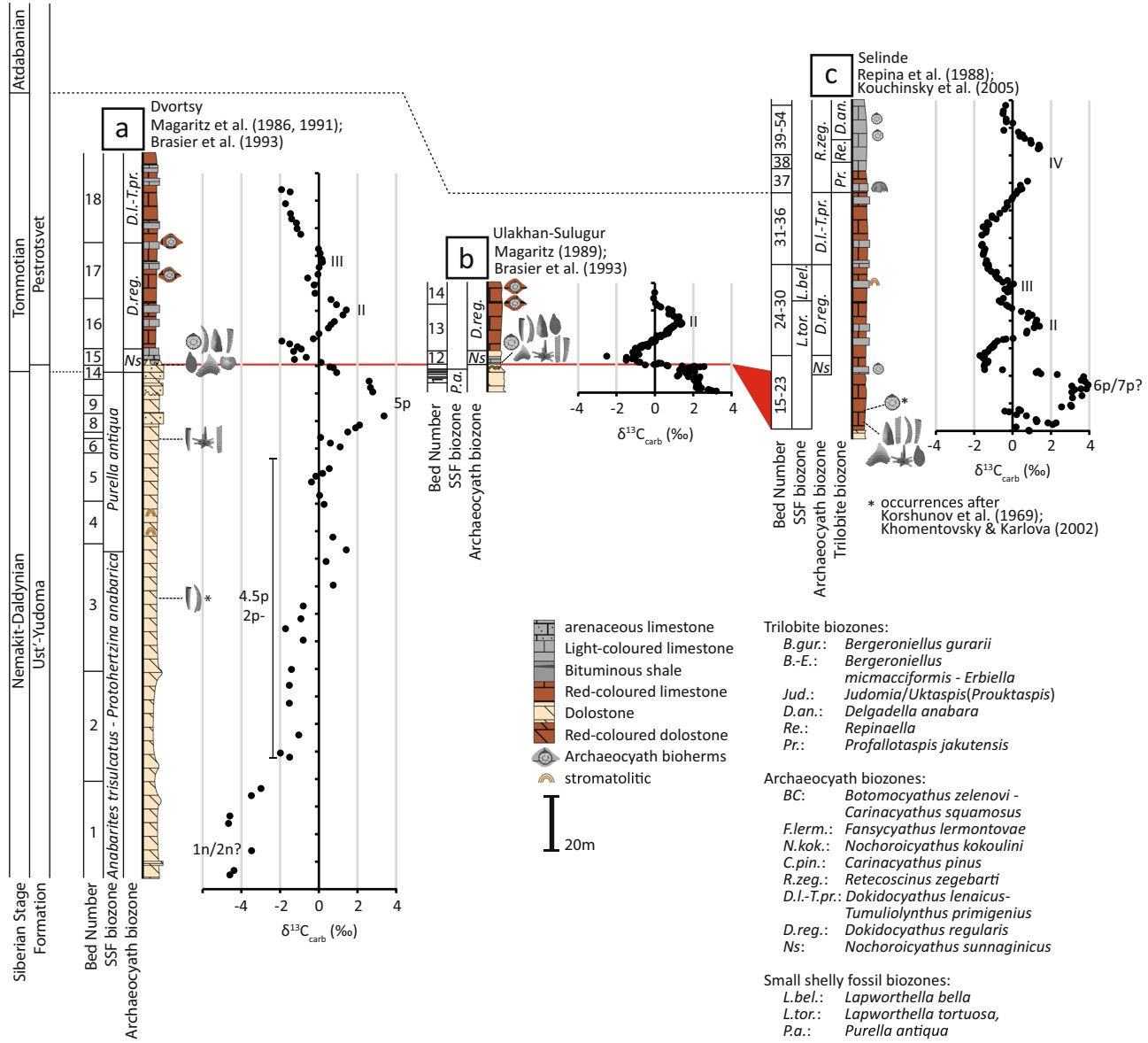

**Fig. 2 | Lithostratigraphy, biostratigraphy and carbon isotope (δ¹³C) data for sections of the Aldan and Selinde rivers bearing the earlierst archaeocyath reef communities of the Siberian Platform. a** Dvortsy[27,28,30] **b** Ulakhan-Sulugur[33,34], and **c** Selinde[69,70].

marine oxia (falling limbs of $\delta^{13}C$ and $\delta^{34}S_{CAS}$), and leading to gradual de-oxygenation over Myr[7]. In addition, phosphorous retention might have occurred under oxic shallow marine conditions, acting to reduce primary productivity and further oxygenate the shallow marine environment in the short-term (<1 Myrs). The OOEs during $\delta^{13}C$ peaks IV and VI–VII occur ~520.5 Ma and ~515 Ma, respectively (Fig. 3b).

The carbonate uranium isotope ($\delta^{238}U$) records from the Sukharikha, Bol'shaya Kuonamka, and Kotuykan rivers of Siberia, the Laolin and Xiaotan sections of South China, and the Oued Sdas section of Morocco[4] are calibrated relative to the $\delta^{13}C$ record and archaeocyath biostratigraphy (Siberia only), and show a consistent pattern with $\delta^{13}C$ and $\delta^{34}S_{CAS}$ (Fig. 3c), whereby decreasing values of $\delta^{238}U$ represent global expansions of anoxic, or more specifically euxinic, conditions conducive to widespread reductant (e.g. pyrite) burial[4]. Values of $\delta^{238}U$ reach a nadir during $\delta^{13}C$ peak 6p, and recover to a more positive mean value during the subsequent terminal Stage 2 to lower Stage 3 interval, which may reflect a gradual transition to a less reducing (or less euxinic) global deep ocean characterised by continued redox stratification[4].

## Reef evolution and habitat occupation

Archaeocyath species diversity on the Siberian Platform increased progressively from their first appearance on the Selinde River in the basal Tommotian *Nochoroicyathus sunnaginicus* Zone (Fig. 2c), when they were represented by only 3 non-reef building species, to reach 50 species by the early Atdabanian *Retecoscinus zegebarti* Zone (Fig. 3d)[11,34]. After a middle Atdabanian decline, raw species diversity increased again to reach a maximum of 60 species in the early Botoman[11]. In parallel, origination rates of archaeocyaths increased in two steps, during the Tommotian–early Atdabanian and middle Atdabanian-early Botoman (Fig. 3e). Archaeocyath extinction rates show peaks in the middle Atdabanian and middle Botoman, the latter being the Sinsk Extinction event which resulted in the near extinction of this group and many other metazoans with calcareous skeletons[35].

In the middle Aldan and Lena rivers' area, and on the northern slope of the Aldan Anticline, the first archaeocyath reefs occur at the base of the Pestrotsvet Formation (basal Tommotian *Nochoroicyathus sunnaginicus* archaeocyathan Zone) (Fig. 1c)[36]. In the Dvortsy and Ulakhan-Sulugur sections (Fig. 2a and b) they were hosted within ooid

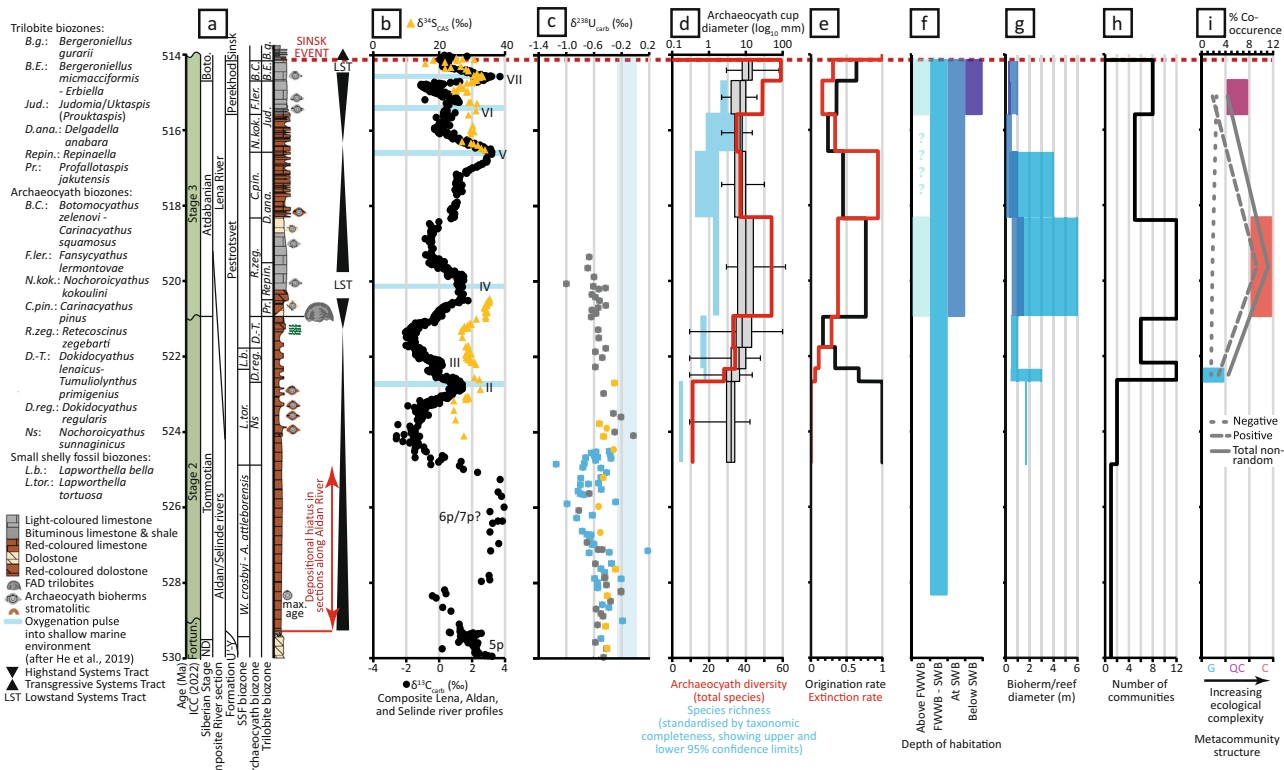

**Fig. 3 | Summary of geochemical and biotic changes through the early Cambrian, Siberian Platform, and uranium isotope data representing a global record. a** International and Siberian timescale, within age model C of [57]. ND−Nemakit-Daldynian regional stage; U'-Y−Ust'-Yudoma Formation. **b** Summary of carbon and sulphur isotopes (from the Lena River, Siberia)[7]. **c** Uranium isotopes from Siberia (grey; Sukharikha and Bol'shaya Kuonamka rivers), South China (blue), and Morocco (orange) (all data points are larger than 2SE)[4]. **d** Archaeocyath sponge species diversity and maximum diameter[12]. Plotted richness values are the species richness estimator[21] with accompanying 95% confidence interval, calculated using the estimated function from R package iNEXT[62]. **e** Rates of archaeocyath sponge species origination and extinction[12]. **f** Reef location as a function of relative water depth (Supplementary Table 1). FWWB−Fair weather wave base. SWB−Storm weather wave base. **g** Reef/bioherm diameter, coloured by relative water depth (see column **f**, and Supplementary Table 2). **h** Number of reef community types (Supplementary Table 3). **i** Archaeocyath reef ecosystem complexity, with percentage of species co-occurrence as changing proportions of total non-random and positive and negative. G = Gleasonian, QG = Quasi-Gleasonian, C = Clementsian.

and skeletal grainstone, and packstone indicating shallow water conditions at or above fair-weather wave base[36,37]. The measured framework of the archaeocyath-calcimicrobial (clotted *Renalcis*) reef is 1.68 m wide and 1.12 m high with the archaeocyath assemblage consisting of 7 species[36].

The basal Pestrotsvet Formation accumulated at the beginning of a transgressive systems tract and deposition continued to the end of the Tommotian (*Dokidocyathus regularis* and *D. lenaicus-Tumuliolynthus primigenius* zones) when patch reefs were widespread across the present Lena River area[38]. Reefs were low biconvex to planoconvex mud mounds, commonly stacked together with sharp boundaries surrounded by peribiohermal sediment, typically argillaceous mudstone and wackestone with skeletal floatstone, rudstone or grainstone. The principal consortia were ramose archaeocyaths (produced by modular branching *Archaeolynthus, Tumuliolynthus* and *Cambrocyathellus* spp.), archaeocyath-*Renalcis* framestone (built by massive modular *Dictyocyathus translucidus* and *Spinosocyathus maslennikovae*), bindstone (composed of plate-like *Okulitchicyathus discoformis*), cementstone, and various archaeocyath-rich mudstones accumulated in subtidal environments between fair-weather and storm-wave base[39,40]. Individual reefs contained between 4 and 20 archaeocyath species including, in places, a distinctive large narrow-conical coralomorph *Cysticyathus tunicatus*[11].

The largest (0.5–2.0 m in length and 0.2–0.5 m in height) and most diverse reefs and community types are found in the *D. regularis* Zone[40]. The belt of reef distribution narrowed in the second part of the zone and reefs disappeared due to regression reducing habitable shelf space by the end of Tommotian 3 (*D. lenaicus-T. primigenius* Zone)[38]. Both individual bioherm size and community type diversity also reduced.

Although scattered archaeocyaths re-appeared at the very beginning of the following Atdabanian Stage (*Retecoscinus zegebarti* Zone), patch reefs only re-appeared towards the end of this zone during a transition from a highstand to the succeeding transgressive tract, and rapidly formed the largest archaeocyath reef belt known on the Siberian Platform, the Oy-Muran reef massif (Fig. 1b and Supplementary Table 2). This reached over 2 km wide along the middle Lena River and was surrounded by mostly argillaceous limestone of the Pestrotsvet Formation. This reef massif was constructed of a number of large low biconvex to planoconvex mud-mounds (1.0–6.0 m in diameter), which like their middle-late Tommotian predecessors, developed on the shallow ramp between fair-weather and storm-wave base and did not produce either prominent palaeorelief or any complex zonation typical of Cenozoic reef systems[41,42]. In addition to *Renalcis*, two larger calcimicrobe species (chambered *Tarthinia* and dendritic *Tubomorphophyton*) participated in framework building. Individual bioherms contained 16–21 archaeocyath species including modular individuals, two coralomorphs (large *Khasaktia vesicularis* bowls and tiny *Hydroconus mirabilis* cones) and a single cribricyath[11].

Archaeocyath buildups also appeared in slightly deeper muddy sediments leeward of the Oy-Muran reef massif where the Negyurchene biohermal massif accumulated (in the vicinity of the Zhurinsky Mys section) (Fig. 1b). Here smaller, lensoid bioherms (about 0.5 m in diameter) grew below storm-wave base within photic zone.

This framework was constructed by branching *Khasaktia* bowls, *Renalcis* and *Epiphyton* with accessory solitary archaeocyaths; rare reefs up to 1.5 m in diameter were formed by clones of branching *Dictyocyathus bobrovi*[11,43]. These reef palaeocommunities consisted of 4–10 archaeocyath species. Further to the West (the Malykan section), ephemeral archaeocyath settlements occasionally formed within agitated intertidal conditions, preserved as grainstones and packstones composed mostly of small fragmented cups of 4 species and the filamentous calcimicrobe *Batinevia*[11]. In total, 12 different archaeocyathan reef palaeocommunities existed during that time and community diversity reached a peak (Fig. 3h and Supplement Data 1).

This reef-building episode terminated at the end of the transgression. Although the Oy-Muran reef massif re-appeared later, severe dolomitization and faulting within this area obscures all palaeobiological detail. Better preserved, however, is the windward area of the transitional facies zone facing the open sea and represented by limestones of the upper Pestrotsvet which are conformably overlain by the lower Perekhod Formation. These accumulated during the following highstand systems tract (from the Bachyk Creek to the Sinyaya River). Here reefs are regularly stalked dendrolites built mainly by *Tubomorphophyton* calcimicrobes with 9–21 species of solitary archaeocyaths restricted to muddy depressions and small primary cavities[11]. These reefs often show a mushroom shape varying in diameter from 0.1 to 0.5 m at different stratigraphic levels. The largest of such bioherms formed wide, laterally contiguous patch reefs or biostromes extending for several tens of square kilometres (e.g. the Bachyk biostrome in Pestrotsvet Formation strata of the *Carinacyathus pinus* archaeocyath Zone). Host sediments of mainly wackestone, rudstone and mudstone have abundant calcimicrobes indicating reef growth near or slightly below storm-wave base within the photic zone. In general, such reefs occurred through the Atdabanian *C. pinus*, *Nochoroicyathus kokoulini* and *Fansycyathus lermontovae* archaeocyath zones. Examples from the *N. kokoulini* Zone strata deposited at the end of a transgressive episode are very rare, but a few ephemeral settlements are present within a muddy floatstone in the Bachyk Creek and some other sections.

Deeper water mud mounds without calcimicrobes up to 0.3 m in diameter, probably developed below storm-wave base and below the photic zone. These occur in the easternmost area of the uppermost Pestrotsvet Formation (*F. lermontovae* Zone in the Ulakhan- and Achchagy-Tuoydakh sections), and were built by 7–12 species of small solitary archaeocyaths and siliceous spiculate sponges surrounded by mudstone[11].

Early Botoman archaeocyaths are known from the upper Perekhod Formation to the East of the Oy-Muran massif and from the Mukhatta Unit to the west of this complex (Fig. 1b). Here rich assemblages are described from reworked perireefal facies represented by oolite and skeletal grainstones. Reefs are known from all relative water depths, reaching 1 m in diameter from above fair-weather base to below storm-wave base. The Sinsk Event, ~513 Ma, represented by the Sinsk Formation, is an inferred anoxic event, and led to a complete disappearance of reef communities during the early Botoman *Bergeroniellus gurarii* trilobite Zone[35]. There was a brief middle Toyonian reef-building episode before final extinction of archaeocyaths on the Siberian Platform[35].

Here we show that individual archaeocyath reef size and extent of habitat, total species diversity, rates of origination and metacommunity complexity increase with phases of oxygenation and decrease as anoxia increases, in pulses, not in a linear fashion, demonstrating patterns of biodiversity and ecosystem integrity were discontinuous, during the Cambrian Radiation.

## Results

In order to assess the relative water-depth zones of reef occupation we categorised the Tommotian 1 to Botoman 1 archaeocyath sponge reef palaeocommunities on the Siberian Platform into seventeen types within eight different environmental settings (SPI–VIII; Supplementary Data 1). We place these eight environmental types within four relative water-depth zones of progressively less energetic and deeper water settings following the early Cambrian palaeogeography of the Siberian Platform[26] and based on associated biota, carbonate lithology and sedimentology as follows (Supplementary Table 1):

### Zone 1 (SPI)
Shallow water archaeocyath skeletal grainstone and packstone formed above fair-weather base at the border of the inner saliniferous basin and the reef belt.

### Zone 2 (SPII–SPIV)
Shallow water reefs hosted within wackestone, rudstone and mudstone, accumulated between fair-weather and storm-wave base within the reef belt.

### Zone 3 (SPV–SPVII)
Deeper water reefs hosted within wackestone, rudstone and mudstone accumulated between near storm-wave base within photic zone within the reef belt and on its margin facing the open basin.

### Zone 4 (SPVIII)
Deep-water reefs hosted within mudstone, deposited below storm-wave base and below photic zone in the proximal part of the open basin.

This analysis shows that reefs were limited to Zone 2 only from the Nemakit-Daldynian to the Tommotian 4 (Fig. 3f). But during the Atdabanian 1 (~521–519 Ma), they notably expanded onshore into Zone 1 and offshore into Zone 3. No data are known from Zone 1 during the Atdabanian 2 and 3, but Zones 2 and 3 remained occupied, and by the Atdabanian 4 and continuing to Botoman 1 (~515 Ma), reefs had extended onshore to Zone 1 and offshore to Zone 4 to occupy all zones.

Archaeocyath species diversity and species richness[12] (standardised by taxonomic completeness, showing upper and lower 95% confidence limits; Fig. 3d) on the Siberian Platform increased in two steps from their first appearance in the basal Tommotian, in the early (~521–519 Ma) and late (~515–514 Ma) Atdabanian (Fig. 3d). A coverage-standardised curve confirms that the pattern of diversity increases/decreases is essentially unchanged once sampling is taken into account (Fig. 3d). Both changing maximum individual archaeocyath skeletal cup diameter and rates of origination broadly follow these trends (Fig. 3d, e; after[12]).

Our reef size analyses are based on compiled reef/bioherm diameter placed within relative water depth zones on the Aldan and middle Lena rivers, Siberian Platform, from Tommotian 1 to Botoman 1 (Supplementary Table 2) shows that reef size was variable (from 0.4 to 3 m) during the Tommotian 1–4. But during the Atdabanian 1–2 (~521–519 Ma), reefs increased to 6 m in maximum diameter in Zone 2, but remained <1.5 m in Zone 3. Reefs were either not present (Zones 1 and 2) or remained small (<1 m; Zones 3 and 4) from the Atdabanian 3 until Botoman 1.

The number of reef environmental community types from Tommotian 1 to Botoman 1 (Supplementary Data 1, Table 3) shows dynamic changes (Fig. 3h). First, a notable increase is observed from the single Nemakit-Daldynian environmental community to the highest number (12) in the Tommotian 2 (~522.5 Ma), then decrease during Tommotian 3 and 4 (~522–521 Ma). A second increase, which occurs during Atdabanian 1 (~521–519 Ma), is followed by a decrease during Atdabanian 2 and 3 (~519–516.5 Ma), and then by a final increase in the Atdabanian 4 and Botoman 1 (~515–514 Ma).

In terms of metacommunity structure, the total archaeocyath species data set exhibits significantly non-random coherence (Z = −45, $p < 0.0001$), positive turnover (Z = 1.05, $p < 0.0001$) and positive

**Table 1 | Metacommunity metrics of co-occurrence, coherence, turnover and boundary clumping, and overall structure type, for the total archaeocyaths species on the Siberian Platform, and metacommunities from the Tommotian 2 (T2), Atdabanian 1 (A1) and Atdabanian 4 (A4)**

| | Co-occurrence | | | Coherence | | | Turnover | | | Boundary clumping | | Structure |
|---|---|---|---|---|---|---|---|---|---|---|---|---|
| | P | N | t-n-r | sM | Z | P-v | sM | Z | P-v | I | P-v | |
| Total | 25.24% | 2.04% | 27.7% | 3597 | −45 | 0.0000 | 262271 | 1.05 | 0.0005 | 1.68 | 0.0000 | Clementsian |
| T2 | 2.87% | 1.64% | 4.50% | 193 | 6.06 | 0.0000 | 874 | 3.11 | 0.0019 | 0.85 | 0.3609 | Gleasonian |
| A1 | 9.35% | 1.87% | 11.20% | 138 | 10.20 | 0.0000 | 968 | 3.36 | 0.0008 | 3.21 | 0.0012 | Clementsian |
| A4 | 1.85% | 2.58% | 4.40% | 318 | 7.29 | 0.0000 | 2587 | 1.07 | 0.2862 | 2.10 | 0.0000 | Quasi-Clementsian |

For Co-occurrence: *P* positive, *N* negative, *t-n-r* total non-random; Coherence and Turnover: *sM* simMean, *Z* Z score, Boundary Clumping: I – Morisita Value; *P-v* P values.

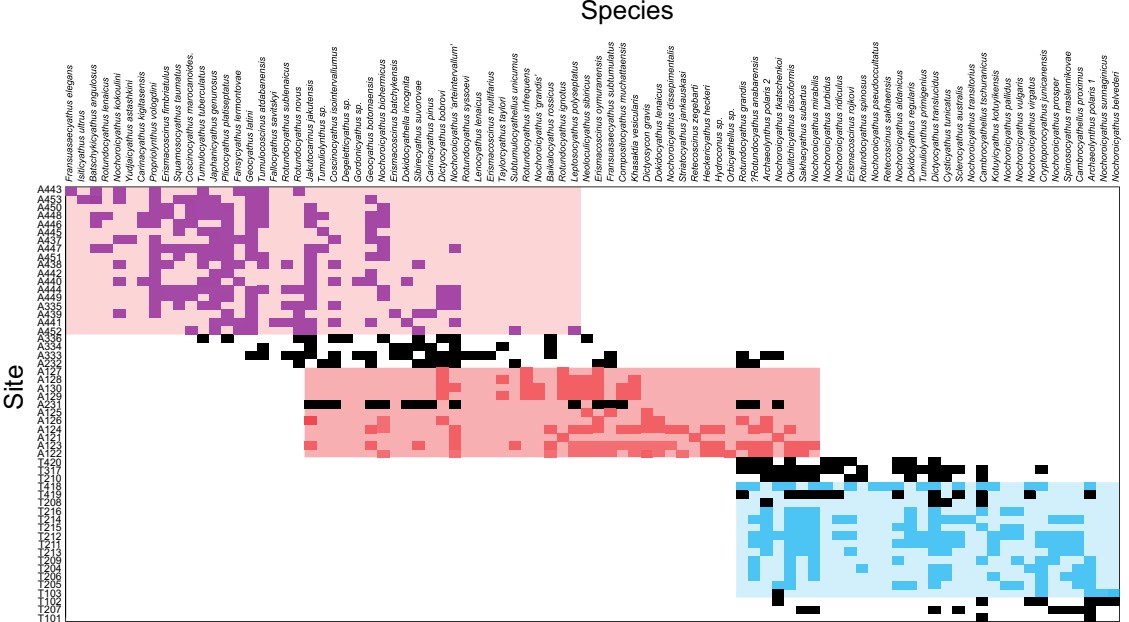

**Fig. 4 | Ordered data used for metacommunity analyses.** Sites (rows) and species (columns) are ordered based on the similarity of community composition, i.e. the number of co-occurring species between sites, as determined by the first step of the metacommunity analyses calculated using reciprocal averaging based on community composition using the R package metacom[63]. The presence of a species is indicated by differently coloured squares which correspond to different time intervals: Tommotian 2 (T2; blue), Atdabanian 1 (A1; red) and Atdabanian 4 (A4; purple). Black squares are sites that do not fit into our T2, A1 and A4 groups, because there are an insufficient number of sites to perform analyses within these time intervals. The least ecologically complex metacommunity (Gleasonian) is indicated by the pale blue background shading, the most complex (Clementsian) by the red-purple background and intermediate complexity (quasi-Clementsian) by the pale red background. The pattern within each of these blocks, rather than patterns within each site (rows), indicates the degree of complexity.

boundary clumping (I = 1.68, *p* = 0.0005), so revealing a Clementsian metacommunity structuring (Table 1; Fig. 4). For this total dataset there was a significant correlation between site ranking and depth (*p* = 0.0002), and a high percentage of significant species co-occurrence of 27.7%, with notably more positive (25.24%) than negative (2.04%) non-random co-occurrences (Table 1).

We compared individual time-binned metacommunities from the Tommotian 2 (~523 Ma), Atdabanian 1 (~521–519 Ma), and Atdabanian 4 (~515 Ma), where sufficient data were available. These show markedly different metrics (Table 1; Figs. 3i, 4). The Tommotian 2 metacommunity exhibited a significantly non-random coherence (Z = 6.06, *p* < 0.0001), positive turnover (Z = 3.11, *p* < 0.0019) and random boundary clumping (*p* = 0.3609), so revealing a Gleasonian metacommunity structure, with low levels of taxa co-occurrence of 4.5%, and similar levels of positive (2.87%) to negative (1.64%) non-random co-occurrences. By contrast, the Atdabanian 1 metacommunity had a significantly non-random coherence (Z = 10.2, *p* < 0.0001), positive turnover (Z = 3.36, *p* = 0.0008) and positive boundary clumping (I = 3.21, *p* = 0.0012), so revealing a Clementsian

structure, with medium levels of taxa co-occurrence (11.2%), and higher number of positive (9.35%) than negative (1.87%) non-random co-occurrences. The Atdabanian 4 metacommunity exhibited significantly non-random coherence (Z = 7.29, *p* < 0.0001), non-significant turnover (Z = 1.07, *p* = 0.2862) and positive boundary clumping (I = 2.1, *p* < 0.0001), so had a Quasi-Clementsian metacommunity structure, with low levels of taxa co-occurrence (4.5%), and a lower number of positive (1.85%) than negative (2.58%) non-random co-occurrences. These patterns of non-random co-occurrences follow those of the paired correlations based on Spearman rank correlations coefficients[11].

Archaeocyath species pairwise associations also change across this interval. None of the species-pairs that occur in both Atdabanian 1 and 4 had significant co-occurrences in either assemblage, but five pairwise species associations changed between Tommotian 2 and Atdabanian 1, and three associations changed from a significant correlation in Tommotian 2, to no significant pattern in Atdabanian 1 (Fig. 4). The total percentage of non-random occurrences follows the same pattern as changing metacommunity type (Fig. 3i).

## Discussion

Reefs have been loci of high biodiversity throughout the Phanerozoic, and so their spatial extent is probably a significant control on changes in marine diversity[44]. Reefs also foster intense competition within their communities for limited resources such as space and food.

Episodes of stable and relatively high seawater temperatures can also stimulate high diversity[45,46]. Indeed, as far as currently quantified, the Siberian Platform maintained the same high temperatures throughout the early Cambrian[9]. Although the Cambrian Radiation proceeded with a marked increase in predation styles inferred to have caused an ecological restructuring of the marine ecosystem[1], such arms races have been suggested to be secondary to the externally-driven climatic stability and productivity factors that create diversity hotspots such as the Siberian Platform[47].

We show that the distribution and size of reefs on the lower Cambrian Siberian Platform follows the two-step increase in archaeocyath diversity, with a notable expansion of reef habitats both offshore to deeper settings and onshore to more shallow settings (Fig. 3f) and an increase in both reef size and archaeocyath sponge species size and diversity during the Atdabanian 1, ~521–519 Ma (Fig. 3d, g), and habitat expansion to even deeper settings by the Atdabanian 4, ~515 Ma (Fig. 3f). The diversity of reef communities follows similar trends (Fig. 3h)[11]. The intervals ~521–519 Ma and ~515 Ma correspond to inferred OOEs during $\delta^{13}C$ peak IV and VI–VII (Fig. 3b)[7].

By contrast, we document decreased archaeocyath sponge body size and diversity (Fig. 3d), increased rates of extinction (Fig. 3e), and a decrease in individual reef size (Fig. 3g) in the intervening interval from the Atdabanian 2 to 3, ~518–516.5 Ma. This corresponds to a prolonged interval of expanding marine anoxia preceding $\delta^{13}C$ peak V (Fig. 3b).

It is archaeocyath sponge individual size, rather than overall morphology (i.e. solitary or modular, branching or encrusting), that correlates with larger reefs within the same facies (Fig. 3d, g), as noted for example in the red argillaceous mudstone of the Pestrotsvet Formation accumulated between fair-wave and storm-wave base. Tommotian and early Atdabanian reefs (~523–519 Ma) developed in the same relative water depth zone, and yet show very different sizes. This may reflect an increase in individual archaeocyath growth rates and longevity[40,48], and we infer that larger reefs were also longer-lived. The diversity of reef communities also follows the pattern of shelf habitat expansion and contraction.

This shift is synchronous with increased diversity and rates of origination, and a quantifiable increase in ecological complexity from Gleasonian to Clementsian communities. The significant increase in positive pairwise species associations between the Tommotian 2 and Atdabanian 1 (~523–519 Ma) metacommunities might indicate changing dynamics within communities with a shift from Gleasonian to more complex and interacting Clementsian metacommunity structures. While increased positive associations may correspond to an increase in suitable mutual habitats, given that the change of metacommunity structure represents an increase in within-community synchronous responses to local environment changes, it is more likely to correspond to an increase in inter-species interactions[15,19]. Such interactions enable more species to co-exist within the same ecosystem, such as mutualisms, thus creating an increase of niche-partitioning, in turn corresponding to an increase in species diversity[11,49,50]. An increase in negative species associations, and drop in positive associations from Atdabanian 1 to 4 (~521–515 Ma) might represent an increase in niche differentiation and/or competitive exclusion.

During the phase of expanded anoxia, ~518–516.5 Ma, we document a decrease in individual reef and archaeocyath sponge size, and diversity, potentially driven by increased rates of extinction. A second oxic pulse in Atdabanian 4 (~515 Ma) shows further habitat expansion and a return to quasi-Clementsian communities.

Hyolith and mollusc species from carbonate settings on the Siberian Platform also show dynamic size trends that are synchronous with archaeocyath sponges, which similarly also positively correlate with increased rates of origination and broadly with total species diversity[12]. The ~521–519 Ma oxic pulse also marks the appearance of trilobites and other metazoan groups on the Siberian Platform (Fig. 3a).

These combined data suggest that oxic pulses resulted in two successive expansion of habitable shelf for reef and other benthos growth. These expansions may have progressed via a deepening of the redoxcline, as a result of shrinking, or deepening of, the OMZ. Available habitat and environmental stability, i.e. habitat age, are established to promote rapid species diversification in reefs[45,47,50,51], so such habitat expansion may have been a major driver in promoting ecosystem complexity perhaps via niche differentiation (Fig. 3f, h, i).

In modern coral reef systems, oxygen availability is among critical factors limiting biodiversity, abundance, calcification rates, colony morphology among others of reef-builders and dwellers[52]. Indeed, this factor is even more important for controlling calcification rates in corals than pH fluctuations[53]. Oxygen availability can influence the growth form of modern sessile benthos including sponges, with domal growth forms dominating under lower oxygen conditions and erect forms within oxic environments[54,55]. Such phenomena are similar to those observed in early Cambrian reefs and related to redox oscillations. A transition from low to higher oxic conditions may therefore have enhanced calcification and growth rates of sessile benthos leading to development of both larger reef inhabitants and reefs themselves, which via various feedbacks promote higher biodiversity, species richness and individual abundance.

Similarly, nutrient input, which controls primary productivity and food availability, can also exert a profound control on the net diversification rate of taxa with higher nutrient availability leading to higher rates[51]. It has been hypothesised that during the lower Cambrian, the long-term burial of reductants that permitted gradual atmospheric (and eventual oceanic) oxygenation, was directly associated with enhanced marine primary productivity[7]. Shallow marine oxygenation would result in sedimentary P retention, limiting primary productivity and acting to expand the oxygenated shelf area. This would be a short-term response to an OOE, coincident with falling limbs of $\delta^{13}C$ excursions. The OOE would reduce the area available for reductant burial, which would eventually result in stifling atmospheric and oceanic oxygenation (coincident with $\delta^{13}C$ nadirs). The long-term burial of reductants ($C_{org}$ and pyrite) during the rising limbs of $\delta^{13}C$ excursions then acts to gradually oxygenate the atmosphere on longer timescales. This scenario then suggests that it was changes in redox, rather than nutrients, that acted as the primary control on the observed changes to ecological metrics, as nutrient limitation due to P retention would be expected during falling $\delta^{13}C$ limbs.

We conclude that oxic pulses may have episodically expanded habitable reef shelf area, each driving biodiversification and metacommunity complexity via community segregation and longevity leading to species diversification and finally niche differentiation. As a result the environmentally stable, tropical lower Cambrian Siberian Platform formed a biodiversity hotspot.

The Cambrian Radiation thus proceeded as a series of 1–3 Myr pulsed ecological and evolutionary metazoan radiations closely following oxygenation events. Yet such phases were short-lived and, importantly, discontinuous and episodic. As a result, we document a highly cyclic timeline, rather than a linear progression, of increasing ecosystem complexity during the Cambrian Radiation. This cyclic timeline is in notable contrast to later Phanerozoic radiations, such as the succeeding Ordovician Biodiversification Event, which progressed within a constantly well-oxygenated shallow marine ocean[56].

## Methods

All our research complies with all relevant ethical regulations. The specimens were collected on the territory of the former Republic of

Yakutia, USSR [ = Republic of Sakha (Yakutia), Russian Federation] during 1978–1987. No special permissions for collecting were required during that period. The specimens were deposited at the Laboratory of Ancient Organisms, Borissiak Palaeontological Institute, Russian Academy of Sciences, Moscow, Russian Federation.

## Correlation

Carbon isotopes have been extensively measured and correlated for all reef sections (Figs. 1, 2)[27–33]. The regional and global correlation of carbon isotope excursions within the Fortunian Stage (corresponding to the regional Nemakit-Daldynian Stage, see Dvortsy section, Fig. 2a) remains poorly calibrated (see discussion in ref. [57]). However, $\delta^{13}C$ excursions in late Stage 2 to Stage 4 are diagnostic and readily correlatable between the Lena and Aldan rivers, and further afield in sections of the northwest (Sukharikha River) and northern Siberian Platform (Bol'shaya Kuonamka River)[58,59]. Indeed, a globally consistent $\delta^{13}C$ chemostratigraphy has been proposed from late Stage 2 onwards that allows integration of $\delta^{238}U$ data from globally distributed sections as shown in Fig. 3c [4,60] (see ref. [57] for age model uncertainties). Details of the radiometric data used to calibrate the resulting $\delta^{13}C$ chemostratigraphy shown in Fig. 3b are provided in ref. [57] and references therein, and additional information used to constrain the upper Atdabanian and lower Botoman interval in ref. [7]. The absolute durations of carbon isotope excursions V–VII remain uncertain.

Data of $\delta^{13}C$ and $\delta^{34}S_{CAS}$ (Fig. 3b)[7], and $\delta^{238}U$ (Fig. 3c)[4] are calibrated within the well-established Siberian biozonation, and all data are subdivided into 0.5–3 Myr time bins following age model C of [57] (Figs. 1, 3b, c), for Cambrian stages 2–3.

## Archaeocyath species diversity and size

Species diversity of archaeocyaths (84 species), coralomorphs (3 species), and cribricyaths (1 species) associated with reefs representing 5930 individuals was derived from all studied reefs (53 sampling units)[11,12]. These species were identified and counted in thin sections and features of their in-situ interactions (e.g. mutual or antagonistic reactions) were taken into account to recognise co-existing species. Species co-occurrences were also identified at the scale of individual thin sections. Between 10 and 50 thin sections were analysed per reef depending on thin section area (2–400 cm²), which were then collated for each reef. Each reef is a single, separate reef (bioherm) with distinct boundaries. The entire sampling set for one site/locality is represented by all the reefs occurring in continuous outcrops at or along the same stratigraphic horizon (Fig. 1b). Temporally separate reefs across the same outcrops are separated by sedimentary rock thickness vertically from 3 to 60 m (Figs. 1c, 2a–c).

Archaeocyath maximum species cup diameter from oriented thin sections was chosen as a measure of body size due to the highly irregular cup and biovolume shape of archaeocyaths[12]. Archaeocyath raw species diversity, and rates of origination and extinction were assembled from[12]. Coverage-standardised species diversity was calculated from[12], as it includes species abundances.

## Archaeocyath species richness

Archaeocyath species richness was estimated (standardised by coverage, showing upper and lower 95% confidence limits) using R statistical software[61], using the estimateD function from package iNEXT[23,62].

## Reef habitat, community type and size

Reef habitat Zone (relative water depth) occupation (Supplementary Table 1) and diversity of reef communities per biozone (Supplementary Data 1 and Table 3) was compiled from[11], and reef/bioherm diameter quantified from the field observations and literature on the Aldan and middle Lena rivers (Supplementary Table 2).

## Metacommunity analyses

Metacommunity analyses were performed in the R package metacom[17], with code provided in the Supplementary Code file. To test for a relationship between depth and community structure, Spearman rank correlations between site scores obtained from reciprocal averaging of the community compositions[17] and the depth were performed on the total data set. There was not enough variability within the site sub-sets to test these for this relationship. Metacommunity analyses were also performed on the Tommotian 2 (T2), Atdabanian1 (A1) and Atdabanian 4 (A4) sub-sets of data. Other time periods did not have sufficient localities to include in these analyses. Co-occurrence analyses were performed in the R package cooccur[19] on each of the T2, A1 and A4 sub-sets of data.

The number of localities and species within sets of metacommunities varies, so absolute values of metrics would not permit comparison. Rather, a Z-score is used which compares the number of standard deviations that the observed metric has from the simulated metric mean. Z-scores measures its distance from the mean of randomisation (simulation mean) as the number of standard deviations (thus making it comparable across metrics with different units)[63,64]. These values are calculated for each of the three metacommunity metrics (coherence, turnover and boundary clumping) to compare their significance. If the observed metric is smaller than the simulated metric mean, then the Z-score is negative; if it is greater, then the Z-score is positive.

Coherence measures the extent to which the taxa within a community react to the same environmental biotic or abiotic variables[15], with positive values reflecting responses to the same environmental variables, and negative values reflecting competitive exclusion between taxa and/or separate niches[18]. If coherence is positive, then this structure can be further investigated through the calculation of turnover and boundary clumping.

Turnover measures the extent to which taxa are replaced within different sites[19], with significantly negative values reflecting less turnover between sites than expected from random. Negative turnover metacommunities have taxa ranges which are nested within each other —i.e. community compositions are a subset of the whole species pool[65]. Positive turnover metacommunities have taxa replacement between sites which is non-nested, i.e. different taxa will occur depending on the underlying environmental variables.

Boundary clumping measures the extent to which taxa ranges cluster at the same sites across the environment[15]. When boundary clumping is greater than one, then taxa ranges cluster together, corresponding to different biomes and the transition between them and when it is less than one this corresponds to an evenly spaced metacommunity[16].

## Reporting summary

Further information on research design is available in the Nature Portfolio Reporting Summary linked to this article.

## Data availability

The authors declare that data supporting the findings of this study were originally published in Ref. [11] and are available within this paper, the Supplementary Data 1 file and Supplementary Tables 1–3. Source data and code to generate all figures can be found in Supplementary Data 1 and Supplementary Code 1.

## Code availability

The authors declare that code supporting the findings of this study are available with the Supplementary Code 1 file.

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

## Acknowledgements

This work was funded by UKRI grant NE/S014756/1 (E.G.M.) and Project NE/T008458/1 (R.W. and F.B.).

## Author contributions

A.Z., R.W., E.M., F.B. and A.P., designed the study, provided materials and resources and wrote the paper. A.Z gathered new data, and E.M. and A.P. analysed the data. F.B. prepared the plots.

## Competing interests

The authors declare no competing interests.
