## [Peer Review File · Nature Communications]

Increases in reef size, habitat, and metacommunity complexity associated with Cambrian Radiation oxygenation pulsesReviewers' Comments:

Reviewer #1:

Remarks to the Author:

This paper is basically a re-analysis of the data presented by Zhuravlev et al. (2015) using the metacommunity concept. The result may be novel, and can eventually be presented in a journal. However, I think the paper is poorly written, and it needs to be greatly improved before publication. Also, the implication of the paper is not well discussed. I suggest rejecting this paper. Details are provided below.

First, the data presentation is not really good.

- The metacommunity concept is not really familiar to paleobiologists, who will be the main reader of the manuscript. I had to read through Presley et al. (2010) paper to understand what the terms such as Clementsian and Gleasonian were. These terms are simply presented in the manuscript without mentioning what they are. It is necessary for the authors to present what these terms are, and why they adopted such a term (and methodology) in the Introduction section.
- In the Results section, the authors just state that their data exhibits certain types of metacommunity structuring without interpreting the data. It is impossible to understand what these statistic values are and what they indicate without an explanation or a reference supporting it. For example, in lines 88-92, the authors wrote: "The total archaeocyath species data set exhibits Clementsian metacommunity structuring (Table 1; Fig. 3), with a significant correlation between site ranking and depth ($p = 0.0002$), and a high percentage of significant species co-occurrence of 27.7%, with notably more positive (25.24%) than negative (2.04%) non-random cooccurrences."
- Although the raw data for the metacommunity analysis that the authors used came from Zhuravlev et al. (2015), this is only briefly mentioned at the end of the manuscript: "We determined the metacommunity structure for for the entire data set, then sub-set to metacommunities T2, A1 and A4, which had a sufficient number of archaeocyath sites on the southern Siberian Platform, to be suitable for analyses¹⁰" (lines 219-222).
- In Fig. 2f, depth data is presented without any reference or reasoning. It is not easy to judge where an ancient reef is inhabited without detailed sedimentological studies, and no reference is provided for this information (in Supplementary Data 1).

Second, possibly more important, is that the manuscript lacks implication.

- Most of the Discussion section (lines 119-140) is more like an interpretation than a discussion, as these sentences only interpret what the result means.
- The implication based on this interpretation is simple; "These combined data suggest that oxic and/or productivity pulses resulted in an expansion of habitable shelf for reef and other benthos growth (lines 141-142)". Such an idea has been suggested numerous times, including a study by the authors (e.g., Zhuravlev and Wood, 2020). How is the result of this study similar or different from previous studies? What do we learn from the result of this study? These were only briefly discussed in lines 141-148. A more lengthy discussion is absolutely necessary here.

Reviewer #2:

Remarks to the Author:

In this manuscript, Zhuravlev et al. present data from archaeocyath reef communities preserved in Cambrian Stage 2-3 strata of the Siberian Platform. On the basis of correlations between ecological metrics of archaeocyath community complexity—including body size, species diversity, nature of co-occurrences, and reef size and environmental distribution—and geochemical indicators of marine redox state, they suggest that oxygenation events directly facilitated increases in complexity and the environmental expansion of early Cambrian reef communities. Additionally, this work corroborates a growing body of evidence indicating that the development of complex seafloor ecosystems across the Neoproterozoic-lower Paleozoic transition was a protracted and episodic process.

This study impressively leverages a large paleontological and sedimentological dataset to shed new light on a critical interval in the rise of complex animal life. These findings will likely be of broad interest to readers across the Earth and life sciences, and I am supportive of publication. My chief suggestion is that the manuscript would be strengthened by additional context and clarification regarding the reef data employed, the framework for the paleocommunity complexity metrics, and integration of the paleontological and geochemical datasets. Some of this information is provided in either the main-text Methods or the Supplement, but in cases the details are still rather sparse and moving some of this to the main text would, I think, enrich and provide clearer support for the authors' key points. I believe the main-text body of Nature Communications manuscripts may be up to 5,000 words; if so, the authors should have sufficient space to expand their manuscript to address these points.

For instance, even though both the archaeocyath data that are the focus of this study and most of the S, C and U isotope data supporting the comparison to changes in paleoredox state are from the Siberian Platform, it was not immediately apparent whether these are all derived from the same sections. From Fig. 1 and 2 and the Supplement it seems as though the He et al. (2019, Nature Geoscience) C and S isotope datasets were generated from localities from which the archaeocyath data were also collected (Lena and Aldan River sections), but the Dahl et al. (2019, PNAS) U isotope dataset is from a different set of Siberian localities. Even where these proxy data have been interpreted to reflect global ocean conditions, it would be useful to clarify, where these are first introduced (e.g., l. 53-67), the origins of each of these records and to make this paleogeographic context more transparent in the main text.

Additionally, it would doubtless be helpful to readers—especially readers who are not experts in paleocommunity analysis—to provide more background regarding the metacommunity structural categories (e.g., Clementsian vs. Gleasonian)—including a more extensive introduction to what these are, why they are appropriate/important metrics for paleocommunity composition, how and at what level co-occurrences were identified (e.g., at the bedding-plane scale? at the facies or stratigraphic unit scale? At the locality scale?), etc. Some of the information currently included the Methods section, or a more succinct version of this, would be more appropriate in the main text, to guide readers' understanding of the Results and Discussion.

One of the metrics the authors employ to gauge ecological 'success' of archaeocyath reefs communities is their paleoenvironmental distribution, in particular the range of inferred paleo-water depths in which individual reef deposits formed, defined in the Supplement in terms of relationship to fair-weather wave base and the inferred depth of the photic zone. However, how these paleo-water depths were assigned and to what portion of the reef they pertain (e.g., reef crest vs. back-reef vs. base of the reef front) was unclear. For a very small (<1 m diameter) individual mound this distinction may be immaterial, but for larger reef tracts (e.g., the Oy-Muran reef belt described on p. 5 of the Supplement) different portions of these could feasibly have occurred in appreciably different water depths. Similarly, how the relationship between water-depth "zones" and the paleo-photoc zone was determined could have been clearly described. Was this based directly on inferred depth? Or presence/absence of associated photosynthesizing organisms such as macroalgae? Although some of these data are derived from previous studies, these criteria are central to the data presented in this study, and should therefore be at least briefly described here.

Lastly, although the authors' focus is on the role of oxygenation as a driver for increases in ecological complexity and diversification, discussion of the potential evolutionary importance of anoxic episodes, beyond their role in restricting diversity and distribution, could be a fascinating addition. For instance, do the authors think that their data support a role for anoxic episodes in promoting the development of new novelties that were implemented as innovations during the oxic events that followed (cf. Wood and Erwin, 2017, Biological Reviews—and subsequently proposed by Wei et al. (2018, Geology; 2020, EPSL) to characterize the Ediacaran-Cambrian of other regions such as South China)? In other words,

could episodic anoxia have actively paved the way for the increased ecological complexity the authors observe during oxic episodes?

Minor comments

l. 113-114: Can you provide references to supporting data/figures in this study and/or to previously published studies in support of this statement?

1. 141: This sentence, along with the brief parenthetical note in l. 40, is the only mention of the potential role of changes in paleoproductivity in shaping these trends. I suggest that this idea either be further developed or removed.

l. 166-169: It would be useful for the authors to describe the methods they used to evaluate archaeocyath size and species diversity in greater detail, beyond references for the source of the data.

Fig. 1: What do the red line and "S.E." indicate in the upper panel? Is this the Sinsk Event?

Fig. 2: Since U isotope data from China and Morocco are also shown in panel c), consider changing the first line of the caption to reflect this broader geographic scope (e.g., "...through the early Cambrian of the Siberian Platform and correlative regions").

Fig. 3: Please include an explanation for the ordering of sites and taxa along their respective axes. Do these reflect first-order sequential relationships (e.g., beyond the coarser separation of Tommotian from Atdabanian sites along the y-axis), or does this reflect post-hoc ordering based upon analytical outputs?

Sincerely,
Lidya Tarhan

Reviewer #3:

Remarks to the Author:

The authors present new bio-geochemical data from sedimentary rocks of the Siberian Platform, which span across the Cambrian time and major biotic turnovers and changes in the earth systems. The bio-data reveal pulsed oxic events and linked with biodiversity of increase and decrease of reef species/habitats, which may therefore invoke episodic oxygenation events that created pulses of evolutionary diversification and enhanced ecosystem complexity.

The major novelty of this study is the careful and detailed bio data that is linked with ocean redox in the basin, which provides an unprecedented picture of the early Cambrian redox and biodiversity in the reefs. It is common practice in geochemistry to test for oxic/anoxic/euxinic conditions in past environments, but no other study to this date has to my knowledge been able to bring a link between oxic anoxic pulses and biodiversity zones "back to life" in the way. We know from the modern ocean that reducing zones that have a limited spatial extent and that they can migrate and expand or shrink in size leading to the biodiversity of reefs and other species, but similar oceanographic processes have not previously been documented in deep time with such detail particularly from the Cambrian. This study will therefore be very useful and be of interest to link the bio and redox part in deep times.

I find the paper very well written and well documented. The stratigraphic correlations are convincing, and the analytical methods as well as the mechanics of the bio-geochemical proxies are described in great details in methods. I therefore only have a few minor comments before publication of the manuscript.

Minor comments:

1. I recommend expanding the word "OOE". For many readers it may not come across easily as an obvious TLA (3 letter acronym). This is an important segment of this paper. The Ediacaran section is relatively longer time period where we saw the OOE. But here, we are looking into relatively shorter

time periods. Hence the recommendation to expand on the word.

2. The youngest age here is around 514 Ma. Spice event is around 494.6 Ma. Which is not that far in time and we have seen global anoxia (See Gill et al., 2011 Nature). I would recommend to add a few insights into how your work leads to the Spice event which is an important aspect of earth's biosphere.

3. Line 57 needs a reference at the end. Also, there is a bit of need to dwell a bit on the S story here. I see a great detail on the U isotope but at a fundamental level S story should be elaborated, if not here, then at least in the SI.

4. We often consider that 542 is when we changed from anoxic to oxic and Cambrian explosion etc... but you use the word OOE several times (for example line 59) where the reader will assume that the Cambrian was fully anoxic and now we are having an oxic events. Please clear these writings and notions so it reads better. But this is overall quite well written.

5. I am not sure about line 64 with peak 6p. Is this required?

6. What is the red line in fig 1.

7. I like fig 2 a lot. But do you think you have the capacity to draw oxic and anoxic zones like we did in Sahoo et al., 2016. Just a suggestion, not mandate.

8. I am missing a fig caption for the Species vs sites figure.

Bes wishes,
Swapan Sahoo.

Reviewer #1 (Remarks to the Author):

This paper is basically a re-analysis of the data presented by Zhuravlev et al. (2015) using the metacommunity concept. The result may be novel, and can eventually be presented in a journal. However, I think the paper is poorly written, and it needs to be greatly improved before publication. Also, the implication of the paper is not well discussed. I suggest rejecting this paper. Details are provided below.

We thank the reviewer for the forthright critic and hope that the revised manuscript provides a more satisfactory analyses where the novelty is more clear.

First, the data presentation is not really good.

- The metacommunity concept is not really familiar to paleobiologists, who will be the main reader of the manuscript. I had to read through Presley et al. (2010) paper to understand what the terms such as Clementsian and Gleasonian were. These terms are simply presented in the manuscript without mentioning what they are. It is necessary for the authors to present what these terms are, and why they adopted such a term (and methodology) in the Introduction section.

Although the terms Clementsian and Gleasonian communities are the basic terms used in ecology and palaeoecology, we now provide an explanation of their meaning in the Abstract and main text.

- In the Results section, the authors just state that their data exhibits certain types of metacommunity structuring without interpreting the data. It is impossible to understand what these statistic values are and what they indicate without an explanation or a reference supporting it. For example, in lines 88-92, the authors wrote: “The total archaeocyath species data set exhibits Clementsian metacommunity structuring (Table 1; Fig. 3), with a significant correlation between site ranking and depth ($p = 0.0002$), and a high percentage of significant species co-occurrence of 27.7%, with notably more positive (25.24%) than negative (2.04%) non-random cooccurrences.”

We agree, and have now added new sections, Metacommunity structure and Metacommunity analysis, to provide the necessary details of the metacommunity analysis, related statistics, and software used for this study.

- Although the raw data for the metacommunity analysis that the authors used came from Zhuravlev et al. (2015), this is only briefly mentioned at the end of the manuscript: “We determined the metacommunity structure for the entire data set, then sub-set to metacommunities T2, A1 and A4, which had a sufficient number of archaeocyath sites on the southern Siberian Platform, to be suitable for analyses¹⁰” (lines 219-222).

The principal sources of data are now emphasised in the revised manuscript. It should be noted that the data base created by Zhuravlev, Wood and Naimark (2015) is critical for the present study, as it is the most comprehensive data base containing site by site data on archaeocyath species composition, number of individuals in sampling sets, and individual sizes (cup diameter).

- In Fig. 2f, depth data is presented without any reference or reasoning. It is not easy to judge where an ancient reef is inhabited without detailed sedimentological studies, and no reference is provided for this information (in Supplementary Data 1).

We agree, and have now added new sections as follows, Relative water-depth zones of reef occupation, Reef evolution and habitat occupation and Reef habitat, community type, and size (Methods), including references for the sedimentology of reefal facies and their distribution across the palaeoprofile of the Siberian Platform.

Second, possibly more important, is that the manuscript lacks implication.

- Most of the Discussion section (lines 119-140) is more like an interpretation than a discussion, as these sentences only interpret what the result means.

- The implication based on this interpretation is simple; “These combined data suggest that oxic and/or productivity pulses resulted in an expansion of habitable shelf for reef and other benthos growth (lines 141-142)”. Such an idea has been suggested numerous times, including a study by the authors (e.g., Zhuravlev and Wood, 2020). How is the result of this study similar or different from previous studies? What do we learn from the result of this study? These were only briefly discussed in lines 141-148. A more lengthy discussion is absolutely necessary here.

The Discussion had now addressed to (1) the influence of redox factors on the size of reef-builders, their diversity, and the entire community structure including a probable order of events leading to increase of their complexity; (2) the absence of any directionality and incremental progress in the development of reef communities during the Cambrian Radiation; and (3) a comparison of modern and early Cambrian reefs under redox stress.

Reviewer #2 (Remarks to the Author):

In this manuscript, Zhuravlev et al. present data from archaeocyath reef communities preserved in Cambrian Stage 2-3 strata of the Siberian Platform. On the basis of correlations between ecological metrics of archaeocyath community complexity—including body size, species diversity, nature of co-occurrences, and reef size and environmental distribution—and geochemical indicators of marine redox state, they suggest that oxygenation events directly facilitated increases in complexity and the environmental expansion of early Cambrian reef communities. Additionally, this work corroborates a growing body of evidence indicating that the development of complex seafloor ecosystems across the Neoproterozoic-lower Paleozoic transition was a protracted and episodic process.

This study impressively leverages a large paleontological and sedimentological dataset to shed new light on a critical interval in the rise of complex animal life. These findings will likely be of broad interest to readers across the Earth and life sciences, and I am supportive of publication. My chief suggestion is that the manuscript would be strengthened by additional context and clarification regarding the reef data employed, the framework for the paleocommunity complexity metrics, and integration of the paleontological and geochemical datasets. Some of this information is provided in either the main-text Methods or the Supplement, but in cases the details are still rather sparse and moving some of this to the main text would, I think, enrich and provide clearer support for the authors' key points. I believe the main-text body of Nature Communications manuscripts may be up to 5,000 words; if so, the authors should have sufficient space to expand their manuscript to address these points.

We thank the reviewer for supportive suggestions. In accordance, we have substantially revised the manuscript to incorporate the principal details of the primary data set, methods, and tools

used. See paragraphs Metacommunity structure, Relative water-depth zones of reef occupation, and Methods.

For instance, even though both the archaeocyath data that are the focus of this study and most of the S, C and U isotope data supporting the comparison to changes in paleoredox state are from the Siberian Platform, it was not immediately apparent whether these are all derived from the same sections. From Fig. 1 and 2 and the Supplement it seems as though the He et al. (2019, Nature Geoscience) C and S isotope datasets were generated from localities from which the archaeocyath data were also collected (Lena and Aldan River sections), but the Dahl et al. (2019, PNAS) U isotope dataset is from a different set of Siberian localities. Even where these proxy data have been interpreted to reflect global ocean conditions, it would be useful to clarify, where these are first introduced (e.g., l. 53-67), the origins of each of these records and to make this paleogeographic context more transparent in the main text.

The data on archaeocyaths and carbon- and sulphur-isotope data sets originated from the same sections are now clearly separated from the Uranium-isotope data derived from other areas of the Siberian Platform and other regions (South China, Morocco). See Geological Setting and Evolution of Redox.

Additionally, it would doubtless be helpful to readers—especially readers who are not experts in paleocommunity analysis—to provide more background regarding the metacommunity structural categories (e.g., Clementsian vs. Gleasonian)—including a more extensive introduction to what these are, why they are appropriate/important metrics for paleocommunity composition, how and at what level co-occurrences were identified (e.g., at the bedding-plane scale? at the facies or stratigraphic unit scale? At the locality scale?), etc. Some of the information currently included the Methods section, or a more succinct version of this, would be more appropriate in the main text, to guide readers' understanding of the Results and Discussion.

The terms Clementsian and Gleasonian communities and related information are now introduced now in the main text. See 'Introductory part' and Metacommunity structure. The level of co-occurrence is identified by statistical methods (e.g. Spearman rank correlation) based on individual archaeocyath count and determination in oriented (mostly perpendicular to reef axis) thin sections (Methods and references therein).

One of the metrics the authors employ to gauge ecological ‘success’ of archaeocyath reefs communities is their paleoenvironmental distribution, in particular the range of inferred paleo-water depths in which individual reef deposits formed, defined in the Supplement in terms of relationship to fair-weather wave base and the inferred depth of the photic zone. However, how these paleo-water depths were assigned and to what portion of the reef they pertain (e.g., reef crest vs. back-reef vs. base of the reef front) was unclear. For a very small (<1 m diameter) individual mound this distinction may be immaterial, but for larger reef tracts (e.g., the Oy-Muran reef belt described on p. 5 of the Supplement) different portions of these could feasibly have occurred in appreciably different water depths. Similarly, how the relationship between water-depth “zones” and the paleo-photic zone was determined could have been clearly described. Was this based directly on inferred depth? Or presence/absence of associated photosynthesizing organisms such as macroalgae? Although some of these data are derived from previous studies, these criteria are central to the data presented in this study, and should therefore be at least briefly described here.

We have now added new sections as follows, Relative water-depth zones of reef occupation, Reef evolution and habitat occupation and Reef habitat, community type, and size (Methods), including references for the sedimentology of reefal facies and their distribution across the palaeoprofile of the Siberian Platform. Such environmental characteristics of modern reefs as reef crest, back-reef and so on are not applicable to early Cambrian reefs of the Siberian Platform, which represent mostly patch reefs that differ only by size and faunal/calcmicrobial composition.

Lastly, although the authors’ focus is on the role of oxygenation as a driver for increases in ecological complexity and diversification, discussion of the potential evolutionary importance of anoxic episodes, beyond their role in restricting diversity and distribution, could be a fascinating addition. For instance, do the authors think that their data support a role for anoxic episodes in promoting the development of new novelties that were implemented as innovations during the oxic events that followed (cf. Wood and Erwin, 2017, Biological Reviews—and subsequently proposed by Wei et al. (2018, Geology; 2020, EPSL) to characterize the Ediacaran-Cambrian of other regions such as South China)? In other words, could episodic anoxia have actively paved the way for the increased ecological complexity the authors observe during oxic episodes?

This is an interesting point, and anoxic episodes may have led to further increases of ecological complexity of reef systems and other benthic communities, but we need additional data sets to be able to test this hypothesis.

Minor comments

l. 113-114: Can you provide references to supporting data/figures in this study and/or to previously published studies in support of this statement?

Provided.

l. 141: This sentence, along with the brief parenthetical note in l. 40, is the only mention of the potential role of changes in paleoproductivity in shaping these trends. I suggest that this idea either be further developed or removed.

Deleted.

l. 166-169: It would be useful for the authors to describe the methods they used to evaluate archaeocyath size and species diversity in greater detail, beyond references for the source of the data.

All necessary details are provided now in Methods.

Fig. 1: What do the red line and “S.E.” indicate in the upper panel? Is this the Sinsk Event?

Yes, it is the Sinsk Event. Clarified.

Fig. 2: Since U isotope data from China and Morocco are also shown in panel c), consider changing the first line of the caption to reflect this broader geographic scope (e.g., “...through the early Cambrian of the Siberian Platform and correlative regions”).

Added.

Fig. 3: Please include an explanation for the ordering of sites and taxa along their respective axes. Do these reflect first-order sequential relationships (e.g., beyond the coarser separation of Tommotian from Atdabanian sites along the y-axis), or does this reflect post-hoc ordering based upon analytical outputs?

Included: The ordering of the sites is based on the metacommunity analyses of reciprocal averaging reposed on the community composition.

Reviewer #3 (Remarks to the Author):

The authors present new bio-geochemical data from sedimentary rocks of the Siberian Platform, which span across the Cambrian time and major biotic turnovers and changes in the earth systems. The bio-data reveal pulsed oxic events and linked with biodiversity of increase and decrease of reef species/habitats, which may therefore invoke episodic oxygenation events that created pulses of evolutionary diversification and enhanced ecosystem complexity. The major novelty of this study is the careful and detailed bio data that is linked with ocean redox in the basin, which provides an unprecedented picture of the early Cambrian redox and biodiversity in the reefs. It is common practice in geochemistry to test for oxic/anoxic/euxinic conditions in past environments, but no other study to this date has to my knowledge been able to bring a link between oxic anoxic pulses and biodiversity zones “back to life” in the way. We know from the modern ocean that reducing zones that have a limited spatial extent and that they can migrate and expand or shrink in size leading to the biodiversity of reefs and other species, but similar oceanographic processes have not previously been documented in deep time with such detail particularly from the Cambrian. This study will therefore be very useful and be of interest to link the bio and redox part in deep times. I find the paper very well written and well documented. The stratigraphic correlations are convincing, and the analytical methods as well as the mechanics of the bio-geochemical proxies are described in great details in methods. I therefore only have a few minor comments before publication of the manuscript.

We thank the reviewer for detailed comments allowing an improvement of our manuscript.

Minor comments:

1. I recommend expanding the word “OOE”. For many readers it may not come across easily as an obvious TLA (3 letter acronym). This is an important segment of this paper. The Ediacaran section is relatively longer time period where we saw the OOE. But here, we are looking into relatively shorter time periods. Hence the recommendation to expand on the word.

OOE is an appropriate term indeed and is used in the revised manuscript.

2. The youngest age here is around 514 Ma. Spice event is around 494.6 Ma. Which is not that far in time and we have seen global anoxia (See Gill et al., 2011 Nature). I would recommend to add a few insights into how your work leads to the Spice event which is an important aspect of earth’s biosphere.

The reference to Gill et al. (2011) revealing a continuation of unstable redox conditions until the late Cambrian is added. But the SPICE event is much younger, and not really relevant to our study.

3. Line 57 needs a reference at the end. Also, there is a bit of need to dwell a bit on the S story here. I see a great detail on the U isotope but at a fundamental level S story should be elaborated, if not here, then at least in the SI.

The necessary details are now provided in Geological Setting and Evolution of Redox.

4. We often consider that 542 is when we changed from anoxic to oxic and Cambrian explosion etc... but you use the word OOE several times (for example line 59) where the reader will assume that the Cambrian was fully anoxic and now we are having an oxic events. Please clear these writings and notions so it reads better. But this is overall quite well written.

This part is changed to show that the Cambrian Radiation developed in a fluctuating but mostly oxic world.

5. I am not sure about line 64 with peak 6p. Is this required?

Yes, this peak (6p) is important to correlate the sections properly and to indicate that archaeocyaths on the Siberian Platform did exist by this (pre-Tommotian) time already (Fig. 2).

6. What is the red line in fig 1.

This line marks the Sinsk Event. This has now been clarified.

7. I like fig 2 a lot. But do you think you have the capacity to draw oxic and anoxic zones like we did in Sahoo et al., 2016. Just a suggestion, not mandate.

Here we prefer to use lines to indicate the precise time of oxic pulses.

8. I am missing a fig caption for the Species vs sites figure.

*Sorry, the caption was indeed too short indeed. The caption is now revised as: **Fig. 5 | Changes in ecosystem complexity as a function of changes in species metacommunity structure and site occupation.** Tommotian 2 (T2; blue), Atdabanian 1 (A1; red) and Atdabanian 4 (A4; purple). The ordering of the sites is based on the metacommunity analyses of reciprocal averaging reposed on the community composition.*

Please note that For the purpose of open access, the authors have applied a Creative Commons Attribution (CC BY) licence to any Author Accepted Manuscript version arising.

Reviewers' Comments:

Reviewer #1:

Remarks to the Author:

The manuscript has been updated significantly, and the authors corrected most of the problems that I pointed out. I am even surprised that the authors didn't write the manuscript like this from the first time. However, I still find many small problems throughout the text, at least some of which can be critical. English needs to be thoroughly checked throughout the text (below I pointed out only the most significant problems). Details are provided below. Therefore, I suggest accepting this paper with a minor revision.

Lines 67-107: Although I asked to detail what metacommunity analysis is, these long sentences do not really fit in the introduction section. Since there is a Methods section at the end of the manuscript, it will be better to move these sentences there and just summarize these in a couple of sentences in the Introduction section and mention that details are mentioned in the Methods section.

Lines 77-82: It is necessary to cite references here, to show readers how exactly Z-scores can be calculated.

Line 109: the entire data set of 11 then => of them 11

Line 129: some evaporites siliciclastics => some evaporites and siliciclastics

Lines 145-147: reference citation required.

Lines 174-175: reference citation required.

Lines 181-184: reference citation required.

Lines 245-249: reference citation required.

Lines 299-302: this sentence lacks a verb.

Lines 302-304: this sentence is grammatically wrong.

Line 308: $I = 1.68''''$

Line 364: delete ,

Lines 366-369: reference citation required.

Lines 369-372: reference citation required.

Fig. 3g. Supplementary Data 2 is cited here, but in the Supplementary material, there is no such file. Or, Supplementary Table 2?

Figs. 4, 5. Please mention in the figure caption that how this graph is generated, and where the raw data is.

Code for analysis is not cited in the method section.

Supplementary Table 1: before $I'n => In?$ Change - (dash) with – (en-dash).

Supplementary Table 2: Please mention whether size refers to the width, height, or both. Change - (dash) with – (en-dash).

Supplementary Table 3: According to Supplementary Data 1, of which the information in Supp. Table 3 calculated from, there is no reef in Namakit-Daldynian and Botoman, as mentioned in the caption. However, the authors listed that there is one reef community in Namakit-Daldynian and 8 in Botoman. Also, it seems to me that the numbers do not match in the other intervals. Please check and correct Fig. 3h accordingly.

In the caption, "Number of reef communities" => "Number of reef community types".

Reviewer #2:

Remarks to the Author:

This revised manuscript has been considerably improved by the authors' expanded description of the sedimentological and paleoenvironmental context of the Siberian Platform archaeocyath reefs that are the focus of this study, as well as their clarification of how archaeocyath diversity and size data were collected and more detailed description of the paleoecological metrics used to infer changes in metacommunity complexity and assess how reef community structure varied with local and global redox state. I continue to think that this study will be of wide interest across the paleontological, ecological and geobiological communities and am supportive of publication.

However, I have a few additional comments, in some cases following on questions I raised in the previous round of the review, as well as some suggestions of areas of the newly expanded text that could, in my view, use further clarification.

I think it would be helpful to readers if the authors were to more transparently state, early in the main text (e.g., on p. 3) whether the paleocommunity analyses/characterization of ecological complexity were performed for only the archaeocyath components of these Siberian Platform archaeocyath-dominated reef communities, or whether all components of the reef assemblages that could be taxonomically identified were included. For instance, did the paleocommunity analyses performed for this study include the hyoliths, helcionelloid mollusks, coralomorphs or calcimicrobes noted in l. 61 and 174-256? These are mentioned in the context of previous studies as well as the facies descriptions and water depth assignments described in the "Reef evolution and habitat occupation" section of the manuscript. But whether they were included in the paleocommunity analyses is not directly stated in the main text, so far as I could determine.

Additionally, in the previous round of review I asked at what scale species-level co-occurrences were identified (i.e., the scale of observation of the 'raw' data employed for the paleocommunity analyses). The authors noted, in their responses-to-reviewers letter, that archaeocyath species determinations and abundances were determined from thin sections (and in the main-text Methods, they state that archaeocyath size and species determinations were made from thin sections; l. 441-443). But does this mean that species co-occurrences were also identified at the scale of individual thin sections (e.g., species were considered to co-occur only if they were identified in the same thin sections? And then these individual thin section data were spatially and temporally collated?)? I understand that determination of z-scores for coherence, turnover and boundary-clumping were used to assign metacommunity types and that spearman's rank correlation was used to assess the relationship between metacommunity factors and paleoenvironment (as the authors noted in their response to my previous comment). But this does not address the more fundamental question of at what spatial (or temporal) scale individual taxa were considered to be "associated" or "co-occur" (the level of association at which the raw data were considered). Toward this end, it would also be helpful if the authors described how "sites" (cf. l. 70, 88ff) were operationally defined—what was considered a sufficient degree of geographic (or paleogeographic) or stratigraphic separation to constitute a distinct "site"? And at what scale were sites or paleocommunities considered to be "temporally different" (e.g.,

l. 116)? Similarly, it would doubtless be helpful to readers if the authors were to state what, for purposes of this study, they consider to constitute a “community” and/or how this was operationally defined—since this is a term that is variably employed by paleontologists (and even ecologists). This is critical context for understanding the authors’ approach for this study, as well as the main-text description of results and discussion. These terms and the scale at which occurrence data were considered should therefore be briefly and straightforwardly defined early in the main text.

Lastly, aspects of the authors’ discussion of productivity dynamics could, in my opinion, benefit from clarification, especially since this aspect of the manuscript appears to have been expanded rather than deleted (contra p. 6 of the response to reviewers document). In particular, there is some confusion regarding the time scale of nutrient-oxygen feedbacks described in l. 147-151, 402-405 and 413. Typically, increased nutrient input is considered to result, on shorter (<1 myr) time scales, in increased anoxia, not increased oxygenation. So why a nutrient pulse would be expected to help “create” an OOE (on the shorter time scales implied by the authors; i.e., prior to a longer-term response to a change in reductant burial flux; cf. l. 148-150) is unclear. The temporal and spatial scales the authors envision should be clarified and framed in terms of the age model resolution for these successions. Similarly, in l. 402-405, the time scale and relative sequence of environmental changes and responses are important here, as increases in productivity are widely considered to lead to increased anoxia in the short term but, if increased productivity results in increased reductant burial, on geologic time scales this should lead to increased oxygenation.

Minor comments

l. 129: Missing “and” between evaporites and siliciclastics?

l. 137-140: Please also note here the resolution of the paleontological and chemostratigraphic data compiled from these sections (were these also collected at a sub-meter scale?).

l. 160: Or potentially simply less euxinic, but still ferruginous and anoxic. Uranium isotope data are most straightforwardly interpreted to reflect the prevalence of euxinic vs. non-euxinic conditions.

l. 195-213: This larger section contains great additional facies and paleoenvironmental detail. However, these two paragraphs seem to be missing paleo-water depth information comparable to that available in the adjacent paragraphs.

l. 294: Is 6 m the maximum or average diameter for Atdabanian 1–2 reef mounds?

l. 313-314: What is the error associated with the time bin age assignments for these successions?

l. 354: Consider replacing “with an ability to build larger reefs” with the less speculative “with larger reefs” (or else provide additional detail and referencing).

l. 408, 415: What does “stability” denote here (if “external” but separate from climate or productivity)? Presumably the authors do not mean redox state?

l. 414: What do the authors envision to have been the mechanism for community segregation?

l. 459: Which subsets of data?

Fig. 3: As noted in the previous round of review, the end of the first sentence of this caption should be changed, given that the data in panel c) are from a broader geographic scale than the Siberian Platform. In panel b) please include error bars for the carbon and sulfur isotope data points (and define the type of error in the caption). Please define the panel c) uranium isotope error bars in the caption, as well as the blue-shaded region. Consider replacing the light sea-green color in panel f) with another color (it is a little challenging to see). And please define the two blue colors used in panel g).

Fig. 4: Please define "iNEXT" and please remove the figure title from the figure (since this is conveyed by the caption, as well as the axis labels).

Fig. 5: It would be helpful to include the full genus names here, or at least standardize the naming scheme used in this figure for individual taxa (some appear to be identified solely by species name, some by "[G. species]" and some solely by genus name).

Sincerely,
Lidya Tarhan

Reviewer #3:

Remarks to the Author:

The revised version addresses most of the issues raised by the reviewers and particularly my comments to make the manuscript better. In the current version, I don't really have any issues to further enhance the manuscript.

Thank you and good luck,

Dr. Swapan K. Sahoo

RESPONSE TO THE REVIEWERS

Reviewer #1 (Remarks to the Author):

The manuscript has been updated significantly, and the authors corrected most of the problems that I pointed out. I am even surprised that the authors didn't write the manuscript like this from the first time. However, I still find many small problems throughout the text, at least some of which can be critical. English needs to be thoroughly checked throughout the text (below I pointed out only the most significant problems). Details are provided below. Therefore, I suggest accepting this paper with a minor revision.

We thank the reviewer for their positive comments, and hope that the revised manuscript provides clear answers to the remaining queries.

Lines 67-107: Although I asked to detail what metacommunity analysis is, these long sentences do not really fit in the introduction section. Since there is a Methods section at the end of the manuscript, it will be better to move these sentences there and just summarize these in a couple of sentences in the Introduction section and mention that details are mentioned in the Methods section.

Paragraphs dealing with the methods of the metacommunity analysis are reorganized in accordance with these comments. General characteristics of the method are kept in Introduction while subsidiary information (former lines 77-97) are now moved to Methods section.

Lines 77-82: It is necessary to cite references here, to show readers how exactly Z-scores can be calculated.

We have now added the following references:

Agarwal, V. & Taffler, J. Twenty-five years of the Taffler Z-score model: Does it really have predictive ability? Account. Bus. Res. 37, 285–300 (2007).

DeVore, G. R. Computing the Z-score and centiles for cross-sectional analysis: A practical approach. J. Ultrasound. Med. 36, 459–473 (2017).

Line 109: the entire data set of 11 then => of them 11

Corrected.

Line 129: some evaporites siliciclastics => some evaporites and siliciclastics

Corrected.

Lines 145-147: reference citation required.

A reference is now added: He et al. (2019)⁷.

Lines 174-175: reference citation required.

A reference is added: Riding & Zhuravlev (1995)³³.

Lines 181-184: reference citation required.

A reference is added: Zhuravlev (1998)³⁵.

Lines 245-249: reference citation required.

A reference is added: Zhuravlev et al. (2015)¹¹.

Lines 299-302: this sentence lacks a verb.

Corrected, a verb is now added.

Lines 302-304: this sentence is grammatically wrong.

This sentence is now rephrased.

Line 308: I = 1.68”,”

Corrected, a comma is added.

Line 364: delete ,

Corrected, the comma is deleted.

Lines 366-369: reference citation required.

References are now added: Leibold & Mikkelsen (2002); Griffith et al. (2016)^{15,19}.

Lines 369-372: reference citation required.

References are now added:

Bode, M., Connolly, S. R. & Pandolfi, J. M. Pleistocene differences drive nonneutral structure in Pleistocene coral communities. Am. Nat. 180, 577–588 (2012).

DiMichele, W. A. et al. Long-term stasis in ecological assemblages: Evidence from the fossil record. Annu. Rev. Ecol. Evol. Syst. 35, 285–322 (2004).

Zhuravlev et al. (2015)¹¹.

Fig. 3g. Supplementary Data 2 is cited here, but in the Supplementary material, there is no such file. Or, Supplementary Table 2?

Corrected – we now refer to Supplementary Table 2.

Figs. 4, 5. Please mention in the figure caption that how this graph is generated, and where the raw data is.

The required captions are added.

Fig. 4. Archaeocyath species richness, standardized by coverage, showing upper and lower 95% confidence limits. Archaeocyath species richness was calculated (standardized by coverage, showing upper and lower 95% confidence limits) using R statistical software⁵⁸, using the estimateD function from package iNEXT^{59,60}.

Fig. 5. Changes in ecosystem complexity as a function of changes in species metacommunity structure and site occupation. Tommotian 2 (T2; blue), Atdabanian 1 (A1; red) and Atdabanian 4 (A4; purple). The ordering of the sites is based on the metacommunity analyses of reciprocal

averaging based on the community composition using the R package *metacom*⁶¹

Code for analysis is not cited in the method section.

This has now been added as a supplementary data file.

Supplementary Table 1: before I'n => In? Change - (dash) with – (en-dash).

Corrected: the stratigraphic position of the lowermost archaeocyath site is calibrated according to $\delta^{13}\text{C}$ peak 6p/7p to fit Figs. 2 & 3. Supplementary tables 1-3 are changed in accordance.

Supplementary Table 2: Please mention whether size refers to the width, height, or both. Change - (dash) with – (en-dash).

The size refers to the width (diameter). Indicated.

Supplementary Table 3: According to Supplementary Data 1, of which the information in Supp. Table 3 calculated from, there is no reef in Namakit-Daldynian and Botoman, as mentioned in the caption. However, the authors listed that there is one reef community in Namakit-Daldynian and 8 in Botoman. Also, it seems to me that the numbers do not match in the other intervals. Please check and correct Fig. 3h accordingly.

In the caption, “Number of reef communities” => “Number of reef community types”.

Supplementary Data 1, Supplementary Table 3 and Fig. 3 caption are now changed in accordance. “Number of reef communities” is replaced by “Number of reef community types” throughout the manuscript files.

Reviewer #2 (Remarks to the Author):

This revised manuscript has been considerably improved by the authors' expanded description of the sedimentological and paleoenvironmental context of the Siberian Platform archaeocyath reefs that are the focus of this study, as well as their clarification of how archaeocyath diversity and size data were collected and more detailed description of the paleoecological metrics used to infer changes in metacommunity complexity and assess how reef community structure varied with local and global redox state. I continue to think that this study will be of wide interest across the paleontological, ecological and geobiological communities and am supportive of

publication.

However, I have a few additional comments, in some cases following on questions I raised in the previous round of the review, as well as some suggestions of areas of the newly expanded text that could, in my view, use further clarification.

We thank the reviewer for her support. Here we added information related to the primary sampling set analysis to indicate how the array of species comprising palaeocommunities was defined. Also, the discussion of productivity dynamics is further clarified.

I think it would be helpful to readers if the authors were to more transparently state, early in the main text (e.g., on p. 3) whether the paleocommunity analyses/characterization of ecological complexity were performed for only the archaeocyath components of these Siberian Platform archaeocyath-dominated reef communities, or whether all components of the reef assemblages that could be taxonomically identified were included. For instance, did the paleocommunity analyses performed for this study include the hyoliths, helcionelloid mollusks, coralomorphs or calcimicrobes noted in l. 61 and 174-256? These are mentioned in the context of previous studies as well as the facies descriptions and water depth assignments described in the “Reef evolution and habitat occupation” section of the manuscript. But whether they were included in the paleocommunity analyses is not directly stated in the main text, so far as I could determine.

Further details of the primary community analysis are now clarified in Introduction (paragraph 3), and what systematic groups were included in the analysis. The number of raw sampling sets and species analysed are now given in the Methods. All these data are provided in Zhuravlev et al. (2015)¹¹.

Additionally, in the previous round of review I asked at what scale species-level co-occurrences were identified (i.e., the scale of observation of the ‘raw’ data employed for the paleocommunity analyses). The authors noted, in their responses-to-reviewers letter, that archaeocyath species determinations and abundances were determined from thin sections (and in the main-text Methods, they state that archaeocyath size and species determinations were made from thin sections; l. 441-443). But does this mean that species co-occurrences were also identified at the scale of individual thin sections (e.g., species were considered to co-occur only if they were identified in the same thin sections? And then these individual thin section data were spatially and temporally collated)? I understand that determination of z-scores for coherence, turnover and boundary-clumping were used to assign metacommunity types and that spearman’s rank

correlation was used to assess the relationship between metacommunity factors and paleoenvironment (as the authors noted in their response to my previous comment). But this does not address the more fundamental question of at what spatial (or temporal) scale individual taxa were considered to be “associated” or “co-occur” (the level of association at which the raw data were considered). Toward this end, it would also be helpful if the authors described how “sites” (cf. l. 70, 88ff) were operationally defined—what was considered a sufficient degree of geographic (or paleogeographic) or stratigraphic separation to constitute a distinct “site”? And at what scale were sites or paleocommunities considered to be “temporally different” (e.g., l. 116)? Similarly, it would doubtless be helpful to readers if the authors were to state what, for purposes of this study, they consider to constitute a “community” and/or how this was operationally defined—since this is a term that is variably employed by paleontologists (and even ecologists). This is critical context for understanding the authors’ approach for this study, as well as the main-text description of results and discussion. These terms and the scale at which occurrence data were considered should therefore be briefly and straightforwardly defined early in the main text.

All the species included in the analysis as well as their co-occurrences and a number of individuals are determined in thin sections under a binocular microscope. The archaeocyath skeletal composition (calcite similar to hosting rock) and their tiny features rarely allow researchers to identification species in the field or in hand specimen.

*Each sample, which was thin sectioned, had been preliminary oriented in the field and then cut longitudinally (parallel to growth axes) and transversally. The entire sampling set represented all the reefs occurring within continuous outcrops. Each sampling set represented a single reef (bioherm) with distinct boundaries (a distinct site), which was spatially separated from another reef by a distance of a few metres to many of kilometres (Fig. 1b). Temporally (across the field section) the reefs were separated by a rock thickness from 3 to 60 m (Figs. 1c-f, 2a-c). The sections were correlated by direct tracing of strata in the field, biostratigraphic methods (trilobite and archaeocyath assemblage zones) and chemostratigraphy ($\delta^{13}\text{C}$ record). The co-occurrence of species was identified for sessile reef-dwellers only (archaeocyaths, a few coralomorphs, and a single cribricyath), whose primary interactions were identifiable in thin sections. Such interrelationships included mutual overgrowth, usually expressed in a deterioration of subdued species, antagonistic escaping and other similar features. See details of palaeoecology of these reefs in Wood et al. (1992, *Functional biology and ecology of Archaeocyatha*. *Palaios* 7, 131–156); Zhuravlev & Wood (Lower Cambrian reefal cryptic communities. *Palaeontology* 38, 443–470) and supplementary references. These is a necessary*

procedure because even shells from a cm-thick layer can represent time-averaging assemblages accumulating fossils for dozens of thousand years rather than a genuine palaeocommunity 'snapshot' (e.g. Kidwell, S. M. Time-averaging in the marine fossil record: overview of strategies and uncertainties. Geobios 30, 977–995).

Lastly, aspects of the authors' discussion of productivity dynamics could, in my opinion, benefit from clarification, especially since this aspect of the manuscript appears to have been expanded rather than deleted (contra p. 6 of the response to reviewers document). In particular, there is some confusion regarding the time scale of nutrient-oxygen feedbacks described in l. 147-151, 402-405 and 413. Typically, increased nutrient input is considered to result, on shorter (<1 myr) time scales, in increased anoxia, not increased oxygenation. So why a nutrient pulse would be expected to help "create" an OOE (on the shorter time scales implied by the authors; i.e., prior to a longer-term response to a change in reductant burial flux; cf. l. 148-150) is unclear. The temporal and spatial scales the authors envision should be clarified and framed in terms of the age model resolution for these successions. Similarly, in l. 402-405, the time scale and relative sequence of environmental changes and responses are important here, as increases in productivity are widely considered to lead to increased anoxia in the short term but, if increased productivity results in increased reductant burial, on geologic time scales this should lead to increased oxygenation.

We agree with the reviewer. This was worded awkwardly in the previous submission, and has now been clarified in the text to make our meaning clearer.

Certainly, nutrient input results in regional marine anoxia on the short-term, and long-term burial of reduced elements leads to a gradual oxygenation of the atmosphere. The OOE's are then a product of shallow marine oxygenation after gradual build-up of atmospheric O₂. This is described in He et al. (2019), who also suggest that sedimentary P retention under oxic shallow to mid-depth conditions (in the wake of an OOE) would result in nutrient limitation for primary productivity, thereby enhancing shallow marine oxygen concentration (approximately coincident with the falling limbs of $\delta^{13}\text{C}_{\text{carb}}$ excursions).

It is also interesting to consider the timescale of this response relative to changes in the $\delta^{13}\text{C}_{\text{carb}}$ curve. If short-term oxygenation during an OOE corresponds to the falling limb, then the falling limbs may be expected to be 'steeper' (of shorter duration) than the rising limbs. Indeed, this is tentatively suggested by the shape of peak V, which has a prolonged rising limb in all sections. We do not explore this observation in the present manuscript due to the uncertainties in $\delta^{13}\text{C}_{\text{carb}}$ age model construction (no absolute radiometric ages are presently

available to robustly constrain the duration of peak V).

Minor comments

l. 129: Missing “and” between evaporites and siliciclastics?

Corrected.

l. 137-140: Please also note here the resolution of the paleontological and chemostratigraphic data compiled from these sections (were these also collected at a sub-meter scale?).

The field sections indeed were sampled with the same sub-meter scale resolution and the composition of reefs communities were subsequently analysed in thin sections.

l. 160: Or potentially simply less euxinic, but still ferruginous and anoxic. Uranium isotope data are most straightforwardly interpreted to reflect the prevalence of euxinic vs. non-euxinic conditions.

We agree with the reviewer, and the text has been clarified accordingly.

l. 195-213: This larger section contains great additional facies and paleoenvironmental detail. However, these two paragraphs seem to be missing paleo-water depth information comparable to that available in the adjacent paragraphs.

Lines 195-199: These are the same reefs of the Dokidocyathus regularis Zone mentioned in the previous paragraph.

Lines 200-213: Palaeodepth data are now added.

l. 294: Is 6 m the maximum or average diameter for Atdabanian 1–2 reef mounds?

This is the maximum diameter; now corrected.

l. 313-314: What is the error associated with the time bin age assignments for these successions?

These ages are approximate following biostratigraphic and chemostratigraphic correlations with sections bearing syngedimentary zircons suitable for age determinations (Bowyer, F. T. et

al. Calibrating the temporal and spatial dynamics of the Ediacaran - Cambrian radiation of animals. Earth-Sci. Rev. 225, 103913 (2022), doi:10.1016/j.earscirev.2021.103913). There is only one zircon U-Pb age (529.7 ± 0.3 Ma) throughout this interval that anchors available $\delta^{13}C_{carb}$ data on the Siberian Platform (Kaufman, A.J., Peek, S., Martin, A.J., Cui, H., Grazhdankin, D., Rogov, V., Xiao, S., Buchwaldt, R., and Bowring, S., 2012, A shorter fuse for the Cambrian explosion?: Geological Society of America Abstracts with Programs, v. 44, p. 326.). However, there are numerous high-resolution U-Pb zircon CA-ID-TIMS ages published from the Anti-Atlas of Morocco (Landing, E., Schmitz, M.D., Geyer, G., Trayler, R.B., and Bowring, S.A., 2020, Precise early Cambrian U-Pb zircon dates bracket the oldest trilobites and archaeocyaths in Moroccan West Gondwana: Geological Magazine, v. 158, p. 219–238.) that anchor a well constrained biostratigraphic and high resolution $\delta^{13}C_{carb}$ chemostratigraphic framework. The chemostratigraphic data from Morocco, in addition to the relative timing of the first appearance of trilobites, can be used to robustly anchor $\delta^{13}C_{carb}$ peak IV on the Siberian Platform. Additional biostratigraphic and radiometric information for age model calibration in this interval are provided in He et al. (2019) (e.g., the use of a radiometric age from Shropshire (Harvey et al., 2011) to constrain the age of the uppermost Atdabanian Fansycyathus lemontovae Archaeocyath Zone). Despite this radiometric calibration, there remains some uncertainty with regards to the precise duration of $\delta^{13}C_{carb}$ excursions (especially V–VII). These uncertainties have been clarified in the revised main text.

l. 354: Consider replacing “with an ability to build larger reefs” with the less speculative “with larger reefs” (or else provide additional detail and referencing).

We agree - the largest archaeocyaths created the largest reefs (Fig. 3d, g).

l. 408, 415: What does “stability” denote here (if “external” but separate from climate or productivity)? Presumably the authors do not mean redox state?

We have now clarified our meaning.

l. 414: What do the authors envision to have been the mechanism for community segregation?

The community segregation means a formation of communities different in their species composition, which, in turn, is a result of competitive exclusion and/or niche partitioning. Which

of these processes was dominated in archaeocyath reefs of the Siberian Platform can only be addressed with further study.

1. 459: Which subsets of data?

We have clarified this, see Lines 487-489:

Metacommunity analyses were also performed on the T2, A1 and A4 sub-sets of data. Other time periods did not have sufficient localities to include in these analyses. Co-occurrence analyses were performed in the R package cooccur20 on each of the T2, A1 and A4 sub-sets of data.

Fig. 3: As noted in the previous round of review, the end of the first sentence of this caption should be changed, given that the data in panel c) are from a broader geographic scale than the Siberian Platform. In panel b) please include error bars for the carbon and sulfur isotope data points (and define the type of error in the caption). Please define the panel c) uranium isotope error bars in the caption, as well as the blue-shaded region. Consider replacing the light sea-green color in panel f) with another color (it is a little challenging to see). And please define the two blue colors used in panel g).

This caption is now changed. Error bars were not provided for the original data in b) which is merely reproduced here. Error bars are present for data shown in c), but all error bars are smaller than the size of data points. All error bars are 2SE, and this has now been clarified in the figure caption.

The colour of the 'Above FWWB' column in panel f) has now been altered to aid clarity. The two blue colours in panel g) correspond with relative water depth colours defined in panel f). This has now been clarified in the figure caption.

Fig. 4: Please define "iNEXT" and please remove the figure title from the figure (since this is conveyed by the caption, as well as the axis labels).

This is clarified now in the caption:

Fig. 4. Archaeocyath species richness, standardized by coverage, showing upper and lower 95% confidence limits. Archaeocyath species richness was calculated (standardized by

coverage, showing upper and lower 95% confidence limits) using R statistical software⁵⁷, using the estimateD function from package iNEXT^{58,59}.

Fig. 5: It would be helpful to include the full genus names here, or at least standardize the naming scheme used in this figure for individual taxa (some appear to be identified solely by species name, some by “[G. species]” and some solely by genus name).

We have now added this where possible, but the generic names are provided for those species only whose generic affinities are established but they lack specific names because they represent new species requiring a formal description.

Sincerely, Lidya Tarhan

Reviewer #3 (Remarks to the Author):

The revised version addresses most of the issues raised by the reviewers and particularly my comments to make the manuscript better. In the current version, I don't really have any issues to further enhance the manuscript.

Thank you and good luck, Dr. Swapan K. Sahoo

We thank the reviewer for the kind acceptance of our revised manuscript.

Reviewers' Comments:

Reviewer #1:

Remarks to the Author:

I have no further comments. The manuscript can be accepted as it is.

Reviewer #2:

Remarks to the Author:

The authors' revisions and responses to reviewer comments have addressed all of my previously raised concerns. I have only a couple minor editorial suggestions (which can be applied under editorial oversight):

l. 107, 283, 374 and elsewhere: temporal ranges should be denoted as either "from [a] to [b]" or "[a]-[b]" (not "from [a]-[b]").

l. 719: presumably the reference to color should be to column f, not column h?

Sincerely,

Lidya Tarhan

Reviewer #4:

Remarks to the Author:

Overall, I think the submitted manuscript reports the result of an interesting study on a topic of wide interest among geo- and bio-scientists. The results are new and noteworthy. Given the constraints of the online journal's unnecessary formatting convention, which minimizes the actual science (i.e., the methods), I found the manuscript well-written and easy to follow. Since the manuscript has already been through at least one round of reviews and the previous reviewers were mainly pleased with the revision, I'll keep my remarks short. I have three comments that should be easily addressed.

First, I found the description of species richness standardization lacking. The description states that richness is coverage standardized and lists an R functions that was used. While this is a mechanical description of how the method was implemented, it fails to describe what coverage standardization is and why it's appropriate to use in this instance. Moreover, it should be made clear if any parameters/options were used with the estimatedD function or if it was implemented using the default parameters.

Second, on line 91 there is reference to a dataset used in reference 11. The sentence fragment reads "...using the entire data set of11 then on a sub-set of metacommunities...". I think this sentence should be rewritten so that it's not referring directly to a superscripted number—this makes the sentence look incomplete at first glance. I am not sure why the standard in-text citation calling out the author and year isn't used—perhaps it violates the journal's conventions. If you don't want to mention the authors of the paper you could try something like: "using an entire previously published data set11 then on a sub-set of metacommunities".

Third, I found figure 5 to be under-described and not very informative. The caption indicates that the blue, red, and purple colors correspond to site age, which leaves me wondering what black indicates. It is not intuitive how complexity is represented on the diagram. The sites are ordered by complexity (presumably from least complex at the bottom to most complex at the top—but readers should not have to presume), but they are colored by age. I think we are supposed to see the linear trend and assume complexity is increasing from the bottom-right to the top left. If this is the case, perhaps reversing the order of species would help make this more intuitive. This way complexity will be

increasing in stratigraphic order from bottom to top. But this still doesn't really show complexity. What is clear is that assemblages from the top of the figure do not overlap in species composition with the assemblages at the bottom. Is there a way of labeling groups of species along the horizontal axis to indicate how complexity is changing? My suggestions for improving this figure are to 1) order the site by time (not complexity) and color the site labels red, blue, purple, and black (or add horizontal bars at stage boundaries), 2) color the grid cells with a color that corresponds to the degree of complexity (i.e., a heat map), 3) reverse order of species along the top to make the trend increasing from bottom to top, and 4) indicate what biological features are changing across the top that correspond to the changing complexity. You may also want to consider that the site and species labels will be too small to read in a figure that's formatted on a published PDF page.

-Noel Heim

RESPONSE TO THE REVIEWERS

Reviewer #1 (Remarks to the Author):

I have no further comments. The manuscript can be accepted as it is.

We thank the reviewer for their recommendation, and their previous comments which have much improved the manuscript.

Reviewer #2 (Remarks to the Author):

The authors' revisions and responses to reviewer comments have addressed all of my previously raised concerns. I have only a couple minor editorial suggestions (which can be applied under editorial oversight):

l. 107, 283, 374 and elsewhere: temporal ranges should be denoted as either "from [a] to [b]" or "[a]–[b]" (not "from [a]–[b]").

l. 719: presumably the reference to color should be to column f, not column h?

We thank the reviewer for noticing these errors, which we have now corrected as suggested on Lines 123, 298, 364,388, 434, and 745.

Reviewer #4 (Remarks to the Author):

Overall, I think the submitted manuscript reports the result of an interesting study on a topic of wide interest among geo- and bio-scientists. The results are new and noteworthy. Given the constraints of the online journal's unnecessary formatting convention, which minimizes the actual science (i.e., the methods), I found the manuscript well-written and easy to follow. Since the manuscript has already been through at least one round of reviews and the previous reviewers were mainly pleased with the revision, I'll keep my remarks short. I have three comments that should be easily addressed.

First, I found the description of species richness standardization lacking. The description states that richness is coverage standardized and lists an R functions that was used. While this is a mechanical description of how the method was implemented, it fails to describe what coverage standardization is and why it's appropriate to use in this instance. Moreover, it should be made clear if any parameters/options were used with the estimateD function or if it was implemented using the default parameters.

We agree with the reviewers suggestion that we need to improve are description of species richness standardization so have expanded on this on Lines 102-117 as follows

"Species richness estimates are highly sensitive to differences in sampling. When comparing species richness of assemblages from several time intervals, it is advisable to standardise sampling across those assemblages to ensure that changes in species richness are not attributable to sampling differences. One approach is to subsample each time interval down to a standardised number of individuals (size-based rarefaction), but this approach can underestimate changes in richness because it tends to sample low-richness assemblages more completely than high-richness ones²¹ (Chao & Jost 2012). Coverage-based rarefaction, where each sample is down-sampled to a standardised level of taxonomic completeness, avoids this potential issue. The coverage of a sample is the proportion of species in the assemblage which are represented in that sample, and it can be estimated by subtracting the proportion of singletons in a sample from 1 (e.g.²²; see also²¹ Chao &

Jost 2012 for details). We used the estimateD function from R package iNEXT²³ (Hsieh et al. 2020) to produce coverage-standardised species richness estimates with 95% confidence intervals, by down-sampling the sampled assemblage from each time interval to match the coverage of the lowest-coverage interval. We did this by setting datatype = “abundance”, base = “coverage” and leaving all other arguments as default.”

Second, on line 91 there is reference to a dataset used in reference 11. The sentence fragment reads “...using the entire data set of 11 then on a sub-set of metacommunities...”. I think this sentence should be rewritten so that it’s not referring directly to a superscripted number—this makes the sentence look incomplete at first glance. I am not sure why the standard in-text citation calling out the author and year isn’t used—perhaps it violates the journal’s conventions. If you don’t want to mention the authors of the paper you could try something like: “using an entire previously published data set 11 then on a sub-set of metacommunities”.

We agree this phrasing is much better so have amended as follows on Lines 91-92:

“using an entire previously published data set¹¹”

Third, I found figure 5 to be under-described and not very informative.

Our apologies that we were not very clear about what Figure 5 represents. It is a visualisation of the metacommunity structure that is the patterns of the presence/absence of taxa between different sites. The complexity of the metacommunities comes from the patterns that we can see in Figure , which is dependent on the sites order. In order to clarify what this Figure represents we have changed the title and the caption substantially, as follows:

“Figure 5: Data used for metacommunity analyses. Sites (columns) and species (rows) are ordered based on the similarity of community composition, i.e. the number of co-occurring species between sites, as determined by the first step of the metacommunity analyses calculated using reciprocal averaging based on community composition using the R package metacom⁶¹. The presence of a species is indicated by different coloured squares which correspond to different time intervals: Tommotian 2 (T2; blue), Atdabanian 1 (A1; red) and Atdabanian 4 (A4; purple). Black squares are sites that do not fit into our T2, A1 and A4 groups, because there are an insufficient number of sites to perform analyses within these time intervals. The least ecological complex metacommunity (Gleasonian) is indicated by the pale blue background shading, the most complex (Clementsian) by the red-purple background and intermediate complexity (quasi-Clementsian) by the pale red background. The pattern within each of these blocks, rather than patterns within each site (rows), indicates the degree of complexity.”

Specific comments:

The caption indicates that the blue, red, and purple colours correspond to site age, which leaves me wondering what black indicates.

The black sites are localities do not fit into our T2, A1 and A4 groups. We have added this to our caption. Lines 746-757.

It is not intuitive how complexity is represented on the diagram. The sites are ordered by complexity (presumably from least complex at the bottom to most complex at the top—but readers should not have to presume), but they are colored by age.

We apologise for the lack of clarity, which hopefully has now been clarified in the new caption. The

sites are not ordered by complexity, but by similarity of taxa within each sites. We have coloured these by age, because these are the sub-sets of the data. If there were no strong temporal signals with community composition, but rather, for example, strong environmental signals then these age-blocks would be mixed up and grouped by environment instead. We have changed the Figure title to clarify this and the caption to explain this.

I think we are supposed to see the linear trend and assume complexity is increasing from the bottom-right to the top left.

This is not the metacommunity pattern we found. We find an increase in complexity from T2 to A1, but then a decrease in complexity from A1 to A4.

If this is the case, perhaps reversing the order of species would help make this more intuitive. This way complexity will be increasing in stratigraphic order from bottom to top. But this still doesn't really show complexity.

We do not wish to reverse the order, since this is not the message we would like people to take away.

What is clear is that assemblages from the top of the figure do not overlap in species composition with the assemblages at the bottom. Is there a way of labelling groups of species along the horizontal axis to indicate how complexity is changing?

The changing complexity comes from within groups of sites, not the taxa themselves. We have tried to indicate the changing complexity of the metacommunities by now adding in coloured blocks behind the groups of taxa. It is the patterns within the blocks that indicate the degree of complexity.

Reviewers' Comments:

Reviewer #4:

Remarks to the Author:

The authors have satisfactorily addressed all concerns raised in my previous review. The only comment I have is that I think "rows" and "columns" are switched in the first line of the Figure 5 caption. That line reads "Sites (columns) and species (rows) are ordered...". In my version of Figure 5, which is in landscape orientation, the sites are rows and species are columns. Otherwise, I have no further comments. This is an excellent study.

-Noel A. Heim

Reviewer #4 (Remarks to the Author):

The authors have satisfactorily addressed all concerns raised in my previous review. The only comment I have is that I think “rows” and “columns” are switched in the first line of the Figure 5 caption. That line reads “Sites (columns) and species (rows) are ordered...”. In my version of Figure 5, which is in landscape orientation, the sites are rows and species are columns. Otherwise, I have no further comments. This is an excellent study.

We thank Reviewer 4 for their help, and have made the changes suggested (Lines 749-750).